# Identification of four biotypes in temporal lobe epilepsy via machine learning on brain images

Yuchao Jiang [1,13] ✉, Wei Li[2,3,13], Jinmei Li[2], Xiuli Li[4], Heng Zhang[5], Xiutian Sima[5], Luying Li[5], Kang Wang[6], Qifu Li[7], Jiajia Fang[8], Lu Jin[9], Qiyong Gong [4], Dezhong Yao [10,11,12], Dong Zhou [2] ✉, Cheng Luo [10,11,12] ✉ & Dongmei An [2] ✉

Artificial intelligence provides an opportunity to try to redefine disease subtypes based on similar pathobiology. Using a machine-learning algorithm (Subtype and Stage Inference) with cross-sectional MRI from 296 individuals with focal epilepsy originating from the temporal lobe (TLE) and 91 healthy controls, we show phenotypic heterogeneity in the pathophysiological progression of TLE. This study was registered in the Chinese Clinical Trials Registry (number: ChiCTR2200062562). We identify two hippocampus-predominant phenotypes, characterized by atrophy beginning in the left or right hippocampus; a third cortex-predominant phenotype, characterized by hippocampus atrophy after the neocortex; and a fourth phenotype without atrophy but amygdala enlargement. These four subtypes are replicated in the independent validation cohort (109 individuals). These subtypes show differences in neuroanatomical signature, disease progression and epilepsy characteristics. Five-year follow-up observations of these individuals reveal differential seizure outcomes among subtypes, indicating that specific subtypes may benefit from temporal surgery or pharmacological treatment. These findings suggest a diverse pathobiological basis underlying focal epilepsy that potentially yields to stratification and prognostication – a necessary step for precise medicine.

Neurology urgently requires a paradigm shift in biology to establish a neotype, based on the shared pathobiological basis, which is a necessary step towards stratified medicine[1]. The redefinition of disease subtypes, grounded in biological mechanisms rather than relying solely on established clinical guidelines, offers significant advantages. By doing so, clinical trials may more effectively recruit a homogeneous population with shared biological characteristics for targeted drug development and interventions. Recent advancements in artificial intelligence (AI)[2], including machine learning applied to brain imaging data, provide robust tools for classifying individuals based on their brain characteristics. This approach to brain subtyping holds great promise in unraveling the underlying pathophysiological mechanisms within disease subsets, ultimately contributing to personalized treatment[3].

Epilepsy is one of the most common and serious disorders in neurology, affecting over 70 million people worldwide[4]. Approximately one-third of individuals with epilepsy exhibit resistance to antiepileptic drug therapy[5]. Surgery is an effective treatment for drug-resistant focal epilepsy such as temporal lobe epilepsy (TLE)[6]. Anterior temporal lobectomy (ATL), encompassing the lateral temporal

A full list of affiliations appears at the end of the paper. ✉e-mail: yuchaojiang@fudan.edu.cn; zhoudong66@yahoo.de; chengluo@uestc.edu.cn; andongmei2010@gmail.com

neocortex and amygdalohippocampal structures, is a well-established surgical approach recommended at grade A for epilepsy surgery[7]. Nevertheless, around 40–50% patients fail to achieve long-term seizure freedom after surgery[8]. This suggests that TLE may comprise distinct subtypes rather than the same entity in all patients, even though they may present relatively homologous clinical picture of seizures and scalp electrophysiological changes. Multiple phenotypes have been proposed from radiologically findings or intracranial electrophysiological findings[9]. However, these clinical classifications may not provide information related to the treatment prognosis. Therefore, the current challenge lies in establishing new methods to identify the distinct subtypes from non-invasive data and to distinguish those who are more likely benefit from surgery or pharmacological treatment.

Magnetic resonance imaging (MRI) is crucial for the diagnosis and treatment of epilepsy, especially when neurosurgical intervention is being considered[10]. Accumulated evidences in TLE suggest that progressive atrophy servers as a crucial structural MRI characteristic. Notably, patients with a higher seizure frequency exhibit a more rapid progression of hippocampal atrophy[11]. Additionally, cortical thinning in certain neocortical regions demonstrates accelerated progression in individuals with a longer duration of illness[12,13]. However, few studies have delved into the underlying spatiotemporal patterns of pathophysiological processes in the brain. Recently, a data-driven mathematical modeling approach successfully estimated the sequence of disease-specific biomarker changes in TLE, providing support for the hypothesis that atrophy progression can be inferred from cross-sectional MRI data[14]. Nevertheless, the complex pathological mechanisms of TLE suggest the existence of multiple biotypes with distinct atrophy progressions. Thus, there is an urgent need for a systematic characterization of spatiotemporal patterns of atrophy progression in TLE.

AI approaches, including unsupervised machine learning techniques, offer powerful tools for subtyping brain diseases[15–17]. However, one major barrier in identifying differential patterns of disease progression (i.e., progression subtypes) is the lack of sufficient longitudinal data across the lifespan of the disease. Recently, a data-driven machine learning algorithm called Subtype and Stage Inference (SuStaIn) was proposed[18]. This algorithm, which relies on cross-sectional observations (e.g., single time-point MRI scans), aims to uncover diverse neurophysiological progression patterns (i.e., SuStaIn trajectories of MRI abnormalities). Once these SuStaIn trajectories are identified, the trained SuStaIn algorithm can assess the degree of association between an individual's MRI data and each trajectory, as well as the corresponding sub-stage of the trajectory (i.e., individualized inference of subtypes)[18]. By employing SuStaIn, recent studies have achieved identification of distinct disease progressions, including tau deposition in Alzheimer's disease[19], gray matter atrophy in schizophrenia[20], and frontotemporal dementia[21]. In our recent study, we applied SuStaIn and uncovered two stable and distinct biological subtypes of schizophrenia, characterized by diverse psychotic profiles and treatment outcomes. These findings suggest that the stratification of individuals based on schizophrenia biotypes holds promise for enhancing diagnosis and prognosis[20].

In this study, our primary objective was to identify distinct trajectories of gray matter atrophy in individuals with TLE using SuStaIn, thereby classifying them to subtypes based on the spatiotemporal patterning of atrophy. Additionally, we aimed to explore differences in neuroanatomical signatures, clinical characteristics, and treatment outcomes among these subtypes. In a 5-year follow-up cohort, we evaluated prediction performance on classifying the subject who achieves seizure freedom after surgery, through a subtype-specific machine learning prediction classifier. This innovative stratification approach highlights the prognostic potential of imaging-based taxonomy, thus informing the design of future clinical trials.

## Results

### Distinct pathophysiological progressions of brain atrophy

Distinct patterns of spatiotemporal progression of brain atrophy have been identified using SuStaIn, based on cross-sectional MRI data from 296 individuals with TLE. Three distinct trajectories of atrophy, labeled as 'trajectory' 1, 'trajectory' 2, and 'trajectory' 3, were observed (Fig. 1a-c). In 'trajectory' 1, the initial regional volume loss was observed in the left hippocampus, followed by the left thalamus, and then extended to the right thalamus and finally to the left entorhinal cortex and cerebral cortex (Fig. 1a). Conversely, in 'trajectory' 2, volume loss began in the right hippocampus, followed by the right thalamus, and then spread to the left thalamus and left hippocampus before affecting the cerebral cortex (Fig. 1b). Lastly, 'trajectory' 3 displayed a cortical-predominant phenotype. It was characterized by initial reduction in the cortex, specifically involving the bilateral middle and superior frontal lobes. Subsequent cortical atrophy was more severe and expanded to other lobes, including the bilateral parietal, occipital, and temporal lobes. Finally, the subcortical regions, the hippocampus and thalamus, were affected (Fig. 1c). The observed differences in the atrophy trajectories across specific brain regions indicate potential phenotypic heterogeneity in the pathophysiological progressions of TLE. We also estimated spatiotemporal trajectories of brain atrophy in a short-term subsample ($n = 148$, mean disease duration = $4.8 \pm 2.7$ years) and a long-term subsample ($n = 148$, mean disease duration = $17.5 \pm 7.4$ years), separately (Supplementary Materials). There was a similar pattern of the three trajectories in the two disease subsample (Supplementary Materials). This suggests that the distinct spatiotemporal patterns of brain atrophy may not be affected by disease progress.

### Subtype-specific neuroanatomical signatures

The SuStaIn approach calculated the probability of each patient belonging to a specific 'trajectory' (Fig. 1d) and further assigned them to a sub-stage within that trajectory. It is important to note that patients who did not exhibit obvious reductions in any ROI were assigned a 'stage=0' by SuStaIn, indicating a 'normal' neuroanatomical signature. The SuStaIn stages showed correlations with z scores, which represent the degree of thickness/volume decrease in patients relative to a healthy population. Specifically, there was a significant correlation between SuStaIn stages and average cortical thickness (Fig. 1e, trajectory 1: $r = 0.599$, $p < 0.001$; trajectory 2: $r = 0.791$, $p < 0.001$; trajectory 3: $r = 0.847$, $p < 0.001$), as well as the volume of the left hippocampus (Fig. 1f, trajectory 1: $r = 0.627$, $p < 0.001$; trajectory 2: $r = 0.577$, $p < 0.001$; trajectory 3: $r = 0.431$, $p = 0.005$). The significant correlation between SuStaIn stages and right hippocampus volume was only found in the 'trajectory' 1 (Fig. 1g, $r = 0.269$, p = 0.013). These findings suggest that the SuStaIn stage may reflect the underlying neurophysiological and pathological processes. Supplementary Table 1 provides ROI-wise correlation coefficients between SuStaIn stages and regional z scores.

By comparing ROI-wise z scores (Fig. 2) between each subtype and the healthy control group, four distinct neuroanatomical signatures were identified as the left hippocampus-predominant signature (subtype 1), the right hippocampus-predominant signature (subtype 2), the cortex-predominant signature (subtype 3), and the 'normal' signature (subtype 4). Compared to the healthy control group, subtypes 1 and 2 exhibited the most severe atrophy in the ipsilateral hippocampus. In subtype 3, gray matter loss was primarily observed in the neocortices. Conversely, subtype 4 showed increased gray matter volume, with the most pronounced enlargement observed in the amygdala (Supplementary Table 2). Furthermore, the bilateral amygdala volume in subtype 4 was larger than in the other subtypes (left, $t = 6.39$, $p < 0.000001$; right, $t = 7.53$, $p < 0.000001$) as well as the healthy control group (left, $t = 7.63$, $p < 0.000001$; right, $t = 7.40$, $p < 0.000001$) (Supplementary Fig. 1). In addition, comparisons

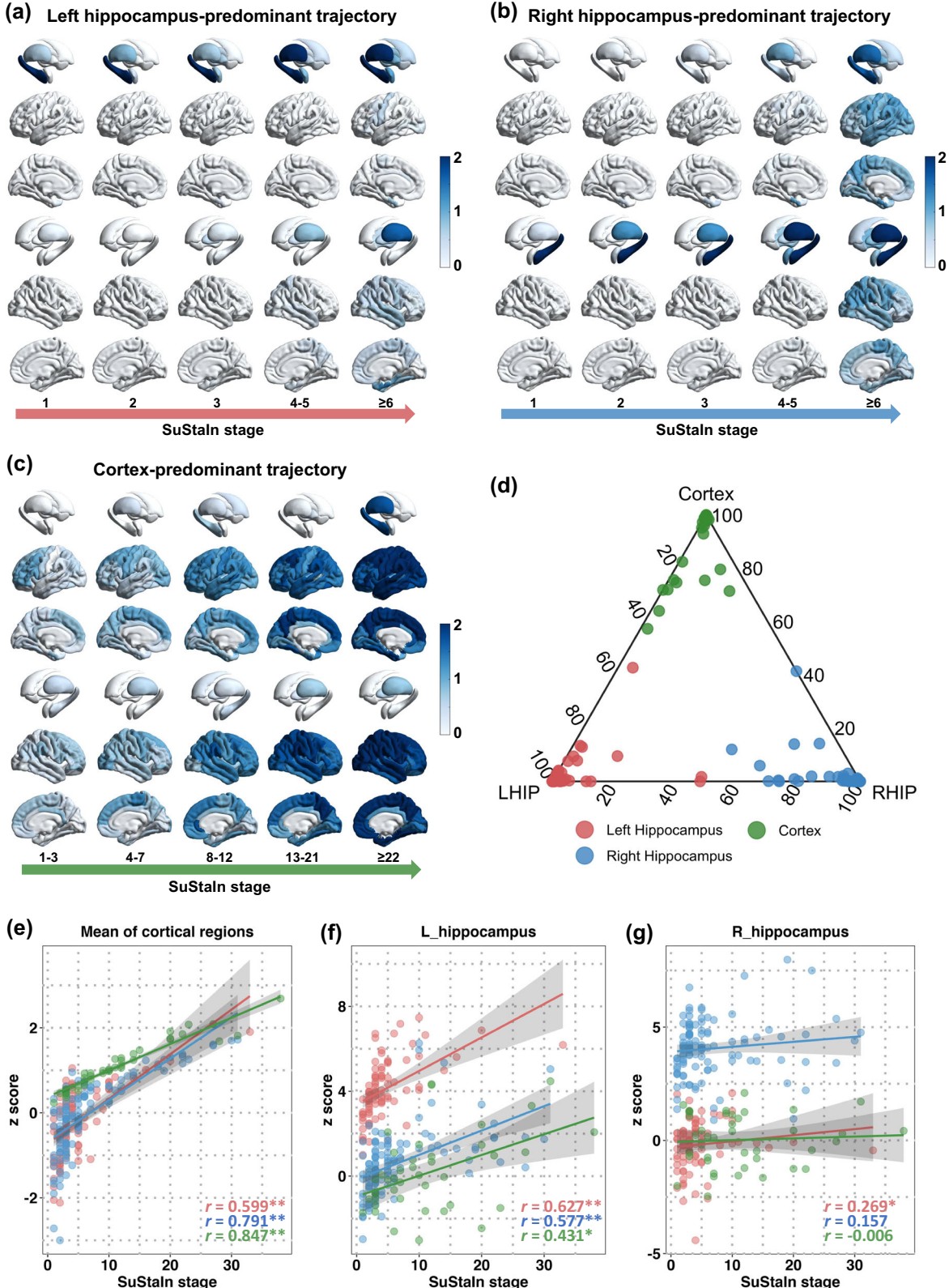

of ROI-wise z score between any two subtypes are visualized in Supplementary Fig. 2. Results of inter-subtype comparison that includes all ROIs across the brain are described in Supplementary Table 3.

## Reproducibility of SuStaIn subtypes

We examined the reproducibility of the SuStaIn trajectories in another independent validation sample including 109 patients diagnosed with temporal lobe epilepsy (61 females, age=33.1 ± 10.4 years). The SuStaIn trajectory was re-estimated based on the validation data. The spatio-temporal trajectory can be mathematically characterized as a sequence of ranked biomarkers (here $n = 23$), which is shown in the (Supplementary Table 4). We observed again that the three trajectories from the validation data were began at the left hippocampus, right hippocampus and cortex separately, consistent with the findings from

**Fig. 1 | Spatiotemporal patterns of progression of brain atrophy via SuStaIn.** Trajectory shows that cortical thickness or volume loss is firstly observed in the left hippocampus (**a**), the right hippocampus (**b**) and cortex (**c**) in people with temporal lobe epilepsy relative to healthy controls. The color of brain region reveals the severity of grey matter loss; white: unaffected areas (z < 1); light blue: mildly affected areas (z = 1–2); dark blue: severely affected areas (z > 2). **d** Individual subtyping according to the maximum probability of belonging to which 'trajectory' (red, left hippocampus-predominant trajectory; blue, right-hippocampus-predominant trajectory; green, cortex-predominant trajectory). **e**–**g** Correlation between SuStaIn stages and z scores (i.e., the degree of thickness/volume decrease in patients relative to healthy population) of average cortical thickness, the volume of left and right hippocampus separately in each subgroup (red, left hippocampus-predominant trajectory; blue, right-hippocampus-predominant trajectory; green,

cortex-predominant trajectory). Spearman correlation test is conducted for data analysis in figures **e**-**g**. It shows a significant correlation between SuStaIn stages and average cortical thickness (trajectory 1: $r = 0.599$, $p = 1.4 \times 10^{-9}$; trajectory 2: $r = 0.791$, $p = 1.8 \times 10^{-25}$; trajectory 3: $r = 0.847$, $p = 3.0 \times 10^{-12}$), as well as the volume of the left hippocampus (trajectory 1: $r = 0.627$, $p = 1.3 \times 10^{-10}$; trajectory 2: $r = 0.577$, $p = 2.3 \times 10^{-11}$; trajectory 3: $r = 0.431$, $p = 0.005$). The significant correlation between SuStaIn stages and right hippocampus volume was only found in the 'trajectory' 3 (trajectory 1: $r = 0.269$, $p = 0.013$; trajectory 2: $r = 0.157$, $p = 0.097$; trajectory 3: $r = -0.006$, $p = 0.973$). The error bands in figures (**e**, **f**, and **g**) represent 95% confidence interval. $n = 85$, 113, and 41 biologically independent samples in left hippocampus-predominant trajectory, right-hippocampus-predominant trajectory, and cortex-predominant trajectory. \*\*$p < 0.001$, \*$p < 0.05$, two-sided. Multiple comparisons were corrected by FDR.

the discovery cohort. In addition, SuStaIn assigned each patient into a subtype, which allowed us to calculate average of z score map across individuals within the same subtype as a representation of subtype-specific atrophy signature. Four distinct signatures of brain atrophy patterning were replicated in the validation dataset (Supplementary Fig. 3). We observed a high consistency of z score map between discovery dataset and validation dataset ($r > 0.7$, $p < 10e{-}10$). These results suggested the reproducibility of SuStaIn subtypes.

In addition, we evaluated the stability of SuStaIn using different number of features. We observed that 93.9% of individuals were consistent for subtype label (Supplementary Table 5), indicating a high stability for individual subtyping even at relatively fewer spatial features for SuStaIn model.

To examine whether the subtype label keeps consistency as disease progresses, we followed up brain MRI data of a subsample ($n = 23$, average of interval time = 39.0 ± 16.8 months). The labels of subtype at follow-up remained consistent with baseline for almost all patients (Supplementary Materials), suggesting that since certain initial brain injury is established, it is less likely to shift from one trajectory pattern (i.e., subtype) to another.

To examine the generalization of SuStaIn subtype to unseen data, we conducted a generalization analysis with ten-fold cross-validation. For each fold, a new SuStaIn model was trained on 90% of the data, and was used to infer individual subtype and stage on the left-out 10% data. We compared whether the subtype and stage assignments of unseen data are consistent with original model that has been trained on all data. We observed that 98.6% of individuals keep consistent subtype assignments with the original subtype (Supplementary Table 6). Spearman correlation test shows a high consistency between stages of unseen data and original result ($r = 0.986$, $p < 0.001$) (Supplementary Materials). These suggest a high generalizability of SuStaIn subtype to unseen data.

### Clinical characterization of subtypes
Among all patients, 28.7%, 38.2%, 13.9%, and 19.2% were classified into subtypes 1 to 4, respectively. Significant differences were observed in various clinical variables among the four subtypes, including age of onset, illness duration, seizure lateralization, MRI hippocampal sclerosis (HS) rate, history of febrile seizures, aura, and treatment outcomes (Table 1). Each subtype exhibited distinct clinical characteristics.

Consistent with expectations, a higher proportion ($\chi^2 = 102.4$, $p < 0.0001$) of individuals with TLE who had positive findings of HS on their MRI were assigned to the left hippocampus-predominant subtype 1 (95.3%) and the right hippocampus-predominant subtype 2 (92.9%) compared to the cortex-predominant subtype 3 (39.0%) and the 'normal' signature subtype 4 (42.1%) (Fig. 3a). Furthermore, the left/right hippocampus-predominant subtypes included patients with TLE whose seizure lateralization was located in the corresponding left or right hemisphere (Fig. 3b), indicating that the initial atrophy occurred primarily in the ipsilateral hippocampus. Patients assigned to the left hippocampus-predominant subtype had the youngest age of onset, with a mean of 12.3 ± 7.7 years, compared to

the other three subtypes ($t = -4.34$, $p < 0.0001$) (Fig. 3c). The left/right hippocampus-predominant subtypes also had a longer illness duration compared to the cortex-predominant subtype 3 and the 'normal' subtype 4 ($t = 3.70$, $p = 0.0003$) (Fig. 3d). We also found a subtype effect on total intracranial volume (TIV) − individuals with the cortical subtype 3 had significantly larger intracranial volume than the other three subtypes; an exploratory analysis was used to examine the association of TIV with subtypes and clinical features.

We investigated whether the neuroanatomical subtype classification based on baseline MRI was related to differential treatment outcomes with medications or anterior temporal lobe surgery. In the medications group (MG, baseline $n = 144$, follow-up $n = 107$), 21 patients reported seizure freedom at the follow-up (mean interval of 56.3 months). In the anterior temporal lobe operative group (OG, baseline $n = 152$, follow-up $n = 145$), 96 individuals reported seizure freedom following the operative treatment at the follow-up (mean interval of 64.1 months). Interestingly, in the medications-treated patients, a significantly higher follow-up seizure freedom rate was observed in the 'normal' signature subtype 4 (39.29%) compared to the other three subtypes (12.66%) ($\chi^2 = 9.29$, $p = 0.0023$) (Fig. 3e). However, for patients treated with anterior temporal lobe surgery, the follow-up seizure freedom rate in subtype 4 was 45.00%, which was significantly worse than the other three subtypes (69.60%) ($\chi^2 = 4.66$, $p = 0.031$) (Fig. 3f).

We also observed that patients with MRI evidence of HS (HS+) show younger age of onset compared to those with normal MRI results (HS-) upon visual examination ($t = -3.49$, $p = 0.001$). In addition, we found that patients with HS- experience worse surgical outcomes compared to those HS+ patients ($\chi^2 = 5.99$, $p = 0.014$). To examine whether the clinical differences among SuStaIn subtypes are affected by HS, we re-analyzed the correlations between clinical features and subtype with HS effect as a covariate (Supplementary Materials). We still found significant correlations of SuStaIn subtype with age of onset ($t = -3.51$, $p = 0.001$), illness duration ($t = -3.15$, $p = 0.002$) and medication outcomes ($\chi^2 = 5.64$, $p = 0.018$) after controlling HS effect (Supplementary Materials).

### Subtype-based classifier predicts surgery prognosis
We evaluated prediction performance on classifying the subject who achieves seizure freedom (OG+) or not (OG-) after surgery, using a classical machine learning prediction procedures (see **Methods**). To examine whether the SuStaIn subtype information could help to improve prediction, we conducted machine learning prediction procedures through a framework under SuStaIn subtype background (Supplementary Fig. 4). We proposed a perspective that each subtype may require specific features/classifiers to predict postoperative outcome, given that each subtype has specific brain structure and clinical characteristics. Thus, using support vector machine (SVM), we built a specific sub-classifier corresponding to each SuStaIn subtype. By ten-fold cross-validation, we observed an acceptable-to-good prediction performance for each sub-classifier to each SuStaIn subtype

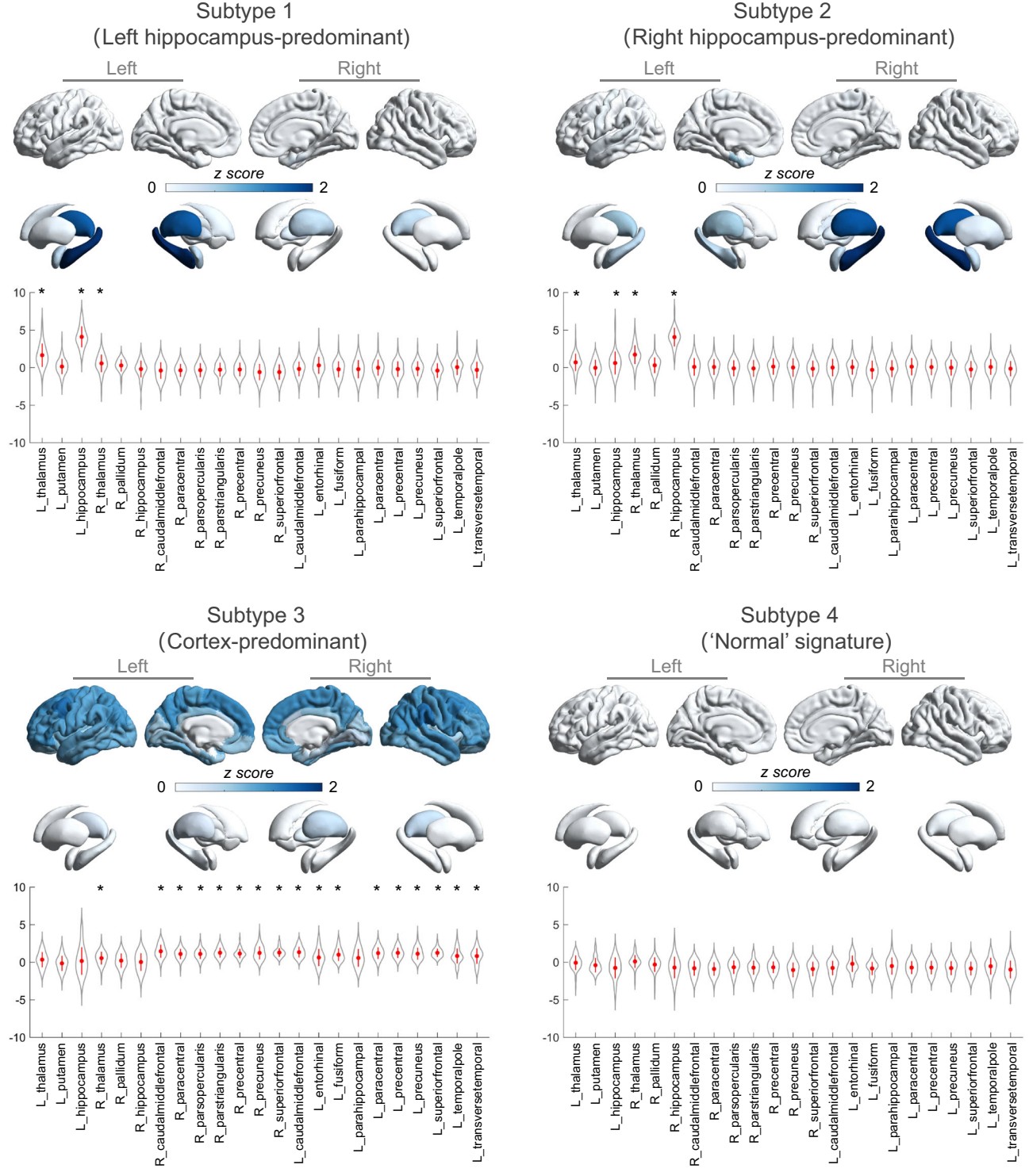

(Supplementary Fig. 5); yielding an overall accuracy (71.72%), specificity (81.03%) and sensitivity (47.87%) on the test data. As a comprehensive evaluation, the Youden Index for the SuStaIn subtype-based classifier ($J = 0.289$) on test data was significantly higher than randomly predictions by permutation test ($p = 0.012$) (Supplementary Fig. 6). Details of prediction performance of each subtype classifier are described in Supplementary Table 7.

As a reference, we also conducted a predictive test without any SuStaIn subtype information as prior. Specifically, SVM classifier was trained using clinical information at baseline as features. By ten-fold cross-validation, we observed 67.59% accuracy, 89.58% specificity and very low sensitivity (24.49%) on the test data; while Youden

Index ($J = 0.141$) did not show significant difference compared to randomly predictions by permutation test ($p = 0.307$). This suggests that these (OG-) patients were not successfully identified if only clinical information was relied upon. In addition, we found that even if we added much more features (clinical variables + MRI regional measures) to train classifier, the prediction performance did not improve (Younden Index = 0.130, accuracy = 66.90%, sensitivity=24.49%, specificity=88.54%).

## Discussion

Our study utilizing a data-driven disease progression modeling algorithm has revealed the presence of phenotypic heterogeneity in the

**Fig. 2 | Four distinct neuroanatomical signatures of brain atrophy patterning in people with temporal lobe epilepsy.** Subtype-specific signature in neuroanatomical pathology includes (1) the left hippocampus-predominant signature (subtype 1), (2) the right hippocampus-predominant signature (subtype 2), (3) the cortex-predominant signature (subtype 3) and (4) the 'normal' signature (subtype 4). ROI-wise z-scores are mapped to a brain template using visualization tools implemented in ENIGMA Toolbox (https://enigma-toolbox.readthedocs.io/en/latest/index.html). Color bar indicates z-scores (i.e., normative deviations) relative to the healthy control group. Note that a higher z-score represents a larger gray matter loss. Data in violin plot are presented as mean values +/− SD. Asterisk indicates significant regional volume reduction in subtype group compared to healthy control group using two-sided two sample $t$-test following FDR multiple comparisons correction. $n = 85, 113, 41$, and 57 biologically independent samples in the subtype 1, subtype 2, subtype 3 and subtype 4. In subtype 1, significant reductions are observed in left hippocampus ($p = 2.9 \times 10^{-43}$), left thalamus ($p = 2.1 \times 10^{-15}$) and right thalamus ($p = 3.0 \times 10^{-5}$). In subtype 2, significant reductions are found in right hippocampus ($p = 7.2 \times 10^{-62}$), left hippocampus ($p = 4.7 \times 10^{-5}$), left thalamus ($p = 3.5 \times 10^{-9}$) and right thalamus ($p = 9.2 \times 10^{-28}$). In subtype 3, significant reductions are found in right thalamus ($p = 7.2 \times 10^{-62}$), right caudalmiddlefrontal (p = $1.8 \times 10^{-13}$), right paracentral ($p = 1.3 \times 10^{-12}$), right parsopercularis ($p = 4.5 \times 10^{-13}$), right parstriangularis ($p = 3.5 \times 10^{-15}$), right precentral ($p = 1.2 \times 10^{-14}$), right precuneus ($p = 2.5 \times 10^{-11}$), right superiorfrontal ($p = 4.2 \times 10^{-17}$), left caudal middle frontal (p = $6.6 \times 10^{-15}$, left entorhinal ($p = 1.2 \times 10^{-3}$), left fusiform ($p = 2.3 \times 10^{-9}$), left paracentral ($p = 2.0 \times 10^{-12}$), left precentral ($p = 3.3 \times 10^{-14}$), left precuneus ($p = 2.8 \times 10^{-11}$), left superiorfrontal ($p = 2.7 \times 10^{-17}$), left temporalpole ($p = 3.9 \times 10^{-6}$), and left transversetemporal ($p = 1.0 \times 10^{-5}$) regions.

## Table 1 | Demographic and clinical characterization of subtypes

| | *n* | Subtype 1 (*n* = 85) | Subtype 2 (*n* = 113) | Subtype 3 (*n* = 41) | Subtype 4 (*n* = 57) |
|---|---|---|---|---|---|
| **Age (year)** | 296 | 25.7(7.8) | 27.8(9.4) | 28.4(6.1) | 27.5(10.4) |
| **Sex (male/female)** | 296 | 43/42 | 58/55 | 21/20 | 35/22 |
| **Age of onset (year)** | 296 | 12.3(7.7)b,c,d | 16(9.7)a,c | 20.1(7.0)a,b | 18.5(11.4)a |
| **Illness duration(year)** | 296 | 13.2(9.1)c,d | 11.8(8.8)c,d | 8.3(6.5)a,b | 8.9(7.0)a,b |
| **Total intracranial volume (cm³)** | 296 | 1456.4(127.8)c | 1472.9(141.9)c | 1617.2(134.6)a,b,d | 1447.8(138.5)c |
| **Seizure lateralization (left/right)** | 296 | 79/6 b,c,d,* | 8/105 a,c,d,* | 21/20 a,b | 36/21 a,b,* |
| **MRI HS rate (%)** | 296 | 95.3% c,d,* | 92.9% c,d,* | 39% a,b,* | 42.1% a,b,* |
| **Handness (right/left)** | 296 | 81/4 | 113/0 | 41/0 | 57/0 |
| **History of hypoxia at birth** | 296 | 4(4.71%) | 5(4.42%) | 1(2.44%) | 4(7.02%) |
| **History of head trauma** | 296 | 7(8.24%) | 11(9.73%) | 2(4.88%) | 8(14.04%) |
| **History of febrile seizures** | 296 | 42(49.41%) c,d,* | 49(43.36%) c,d,* | 9(21.95%) a,b,* | 7(12.28%) a,b,* |
| **History of encephalitis meningitis** | 296 | 15(17.62%)d | 20(17.70%)d | 4(9.76%)d | 0(0.00%)a,b,c,* |
| **History of positive family** | 296 | 1(1.18%) | 5(4.42%) | 2(4.88%) | 6(10.53%) |
| **Aura** | 296 | 62(72.94%)d | 83(73.45%)d | 28(68.29%) | 30(52.63%)a,b,* |
| **Seizure frequency (daily/weekly/monthly/ yearly)** | 296 | 8/35/37/5 | 9/46/46/12 | 2/18/18/3 | 13/24/18/2 |
| **Seizure type (FS/FBTCS)** | 296 | 25/60 | 39/74 | 13/28 | 24/33 |
| **Medications (1/2/3/4)** | 296 | 28/33/21/3 | 33/54/21/5 | 16/15/10/0 | 20/22/14/1 |
| **Pathology waves (unilateral/bilateral)** | 296 | 67/18 | 88/25 | 28/13 | 41/16 |
| **Treatments (OG/MG)** | 296 | 55/30 | 57/56 | 19/22 | 21/36 |
| **MG follow-up (Effective/ineffective/lost)** | 144 | 2/20/8 | 5/33/18 | 3/16/3 | 11/17/8 |
| **MG seizure-free rate (%)** | 107 | 9.09%d | 13.16%d | 15.79% | 39.29%a,b,* |
| **MG follow-up interval(months)** | 107 | 53.4(27.6) | 52.5(26.5) | 62.1(33.7) | 59.7(33.2) |
| **OG follow-up (Effective/ineffective/lost)** | 152 | 36/16/3 | 39/15/3 | 12/7/0 | 9/11/1 |
| **OG seizure-free rate (%)** | 145 | 69.20% | 72.2%d | 63.20% | 45%b,* |
| **OG follow-up interval (months)** | 145 | 64.6(33.1) | 63.1(27.8) | 62.9(30.1) | 66.3(26.6) |

*Corrected two-sided *P* < 0.05 (versus all other subtypes); ªCorrected two-sided *P* < 0.05 (versus subtype 1); ᵇCorrected two-sided *P* < 0.05 (versus subtype 2); ᶜCorrected two-sided *P* < 0.05 (versus subtype 3); ᵈCorrected two-sided *P* < 0.05 (versus subtype 4). ANOVA with post-hoc Least Significant Difference tests is used for continuous variables (age, age of onset, illness duration and total intracranial volume). Pearson's Chi-square test is used for other categorical variables. Multiple comparisons are corrected by FDR. Subtype1, the left hippocampus-predominant signature; Subtype2, the right hippocampus-predominant signature; Subtype3, the cortex-predominant signature; Subtype4, the normal signature; HS, hippocampal sclerosis. FS focal seizure, FBTCS focal to bilateral tonic–clonic seizure, OG operative group, MG medication group.

pathophysiological progressions of TLE. We have identified three distinct trajectories of atrophy progression, highlighting the predominance of hippocampal involvement in the ipsilateral hemisphere and a cortex-predominant phenotype that initiates in the frontal lobe. Based on the temporal heterogeneity observed within these trajectories, we have further categorized a total of four subtypes. These subtypes exhibit differences in their neuroanatomical signatures, clinical characteristics, and long-term treatment outcomes, both in surgical interventions and pharmacological treatments. These findings provide evidence for the biological plausibility of distinct subtypes in TLE and suggest their therapeutic relevance and potential prognostic value. The identification of these biotypes opens avenues for personalized treatment approaches in TLE, facilitating improved patient care and outcomes.

Three diverse trajectories of brain atrophy were identified via imaging-based machine learning, indicating possible origins of neuroanatomical pathology in TLE. Two of the trajectories exhibit a hippocampus-predominant pattern, showing a similar spatiotemporal progression that begins with atrophy in the ipsilateral hippocampus, followed by the ipsilateral thalamus, contralateral thalamus, and other associated structures. This suggests the involvement of both the ipsilateral hippocampus and bilateral thalamus in the pathological propagation of TLE[22]. The findings of our study align with existing knowledge regarding the crucial role of the thalamus in the propagation of epileptic discharges[23] and its involvement in cortical connections during focal to bilateral tonic-clonic seizures (FBTCS)[24]. Furthermore, our study has revealed a cortex-predominant trajectory characterized by delayed hippocampal atrophy compared to the

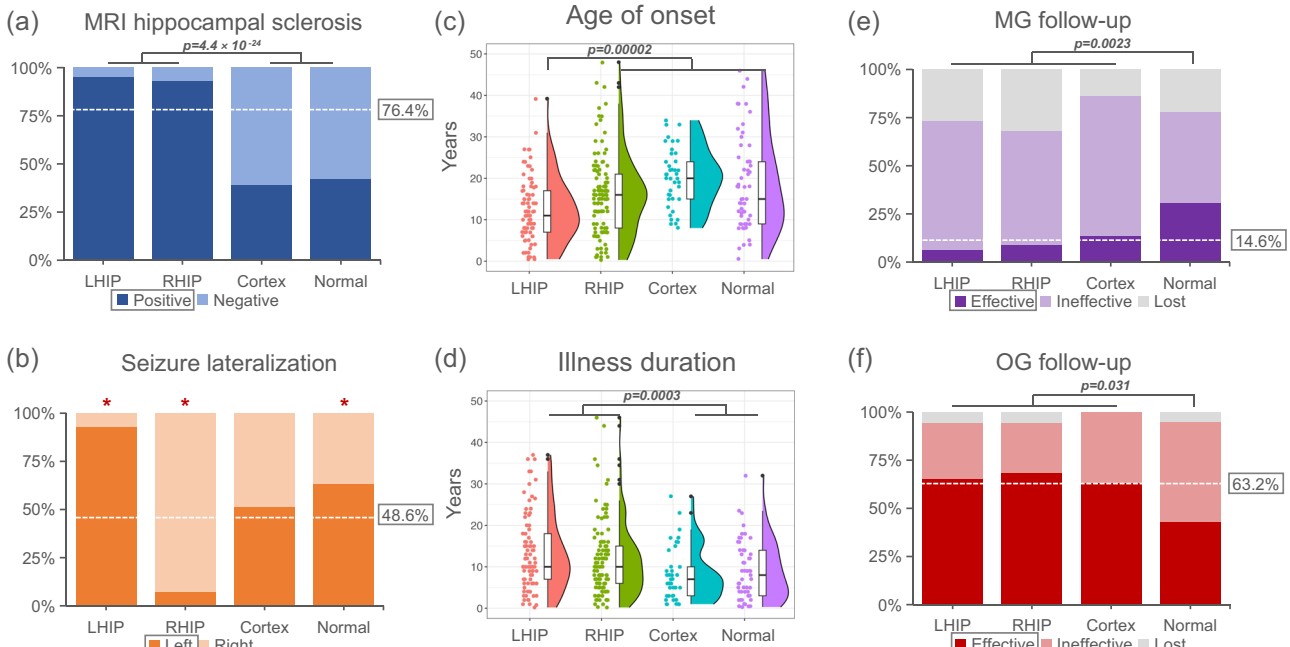

**Fig. 3 | Clinical characterization of subtypes. a** Proportion of TLE individuals with a visible hippocampal sclerosis on their magnetic resonance imaging (MRI) in each subtype. **b** Proportion of individuals with TLE whose seizure lateralization located at the corresponding left or right hemisphere. Red asterisk represents significant difference between a specific subtype vs. all other subtypes (subtype 1, $p = 3.8 \times 10^{-22}$; subtype 2, $p = 2.5 \times 10^{-29}$; subtype 3, $p = 0.723$; subtype 4, $p = 0.015$). **c** Differences of age of onset among four subtypes. **d** Differences of illness duration among four subtypes. **e** Proportion of individuals with seizure-free (i.e., effective), not seizure-free (i.e., ineffective) or lost follow-up in 144 medicated individuals (MG) at the follow-up (mean interval is 56.3 months). **f** Proportion of individuals with seizure-free (i.e., effective), not seizure-free (i.e., ineffective) or lost follow-up in 152 anterior temporal lobe operative individuals (OG) at

follow-up (mean interval is 64.1 months). The white dotted line (a, b, e, and f) shows the average of the four subtypes. Data in figures (**c** and **d**) are presented using a box-plot (center line, median; box limits, upper and lower quartiles; whiskers, 1.5×interquartile range [IQR]; points, outliers). $n = 85$, 113, 41, and 57 biologically independent samples in the subtype 1, subtype 2, subtype 3 and subtype 4. Pearson's Chi-square test is conducted for data analysis in figures **a**, **b**, **e** and **f**. Two-sided two-sample t test is used for data analysis in figures c and d. Multiple comparisons were considered with FDR correction. LHIP, left hippocampus-predominant signature (subtype1); RHIP, right hippocampus-predominant signature (subtype2); Cortex, the cortex-predominant signature (subtype3); Normal, the 'normal' signature (subtype4).

neocortex. This highlights the potential heterogeneity in the temporal sequence of neurodegenerative processes within the cortex and hippocampus. Importantly, we have established a strong correlation between the stage of the trajectory and the extent of regional atrophy, indicating that the trajectory derived from cross-sectional imaging data reflects the underlying pathophysiological progression, particularly related to neurodegeneration[18–20]. Taken together, these spatiotemporal patterns of brain atrophy trajectories provide direct structural imaging evidence supporting the existence of phenotypic heterogeneity in the pathophysiological progressions of TLE. They enhance our understanding of the diverse mechanisms involved and suggest the presence of distinct subtypes within the TLE population.

Based on the individual variability in the spatiotemporal patterns of brain atrophy progression, we have identified four distinct biotypes of TLE, each displaying unique neuroanatomical signatures. The first two subtypes, characterized by "early-occurred" hippocampal atrophy, exhibited a high percentage (over 90%) of patients with evidence of HS on MRI and seizure focus localized to the ipsilateral hemisphere. Notably, these subtypes (subtype1 and subtype2) showed relatively high rates of effectiveness (69.2% and 72.2%, respectively) in ATL surgery. This suggests that surgical intervention yields favorable long-term outcomes for patients with "early-occurred" hippocampal atrophy as the primary signature. Interestingly, the third subtype, characterized by "late-occurred" hippocampal atrophy, displayed a good surgical effectiveness rate (63.2%), despite a lower proportion of patients showing MRI evidence of HS (39.0%). This finding suggests that ATL surgery can be beneficial for the subtypes 1, 2, and 3,

regardless of whether hippocampal atrophy occurs early or late. The latter may be related to progression of epileptogenic neuroanatomic pathology in TLE[12,25]. It is important to note that the underlying mechanisms behind the effectiveness of ATL surgery in these subtypes, particularly in cases with late-occurring hippocampal atrophy and without clear MRI evidence of HS, are not fully understood. Further research is needed to elucidate the specific factors contributing to the surgical outcomes in these cases and to explore the progression of epileptogenic pathology in TLE. For subtype 4, brain morphometry analysis revealed no cortical thickness or subcortical volume reductions in any regions compared to healthy controls. Intriguingly, subtype 4 exhibited a significant increase in amygdala volume relative to both the healthy group and the other subtypes. This finding aligns with previous proposals that amygdala enlargement represents a distinct subtype of TLE[26,27]. This particular subtype is characterized by an older age of epilepsy onset, a greater tendency to nonconvulsive seizures, and a favorable response to antiepileptic drugs[28]. Consistent with these prior observations, our study demonstrated a moderate response (39.3%) to antiepileptic drugs specifically within subtype 4. While the underlying epileptogenic structures or spreading mechanisms associated with amygdala enlargement and its correlation with positive responses to antiepileptic therapies remain unclear[26], our research uses an imaging-based taxonomy, showing divergent long-term responses to antiepileptic treatments.

The SuStaIn subtypes show similar clinical characteristics with four known TLE types (TLE with left HS, TLE with right HS, TLE with negative MRI, and TLE with enlarged amygdala)[27,29]. The neuro-structural features of SuStaIn subtypes were mostly associated with

presence and location of MRI HS[6]. This again confirms the brain structural heterogeneity within individuals diagnosed with TLE. In clinical studies, TLE with HS generally have a worse response to medications and experience seizures at a younger age compared to those with normal MRI upon visual examination[30,31]. This is also observed in SuStaIn subtype 1, which includes most of patients with left side HS. Additionally, previous studies reported that those individuals without hippocampal atrophy usually experience worse responses to surgical treatment[32,33]. In our study, we also observed that patients with HS- experience worse surgical outcomes compared to those HS+ patients. We suspect that this may be one of the reasons for the poor surgical outcomes of SuStaIn subtype 4. In short, the differences between data-driven subtypes are in part consistent with known clinical features.

Subtype is a prior important information to aid prediction of surgery outcomes. We built a classifier cluster including specific subclassifier corresponding to each subtype, which achieved an acceptable-to-good performance on predicting seizure freedom subjects after surgery, better than clinical information-based only prediction model. Although the underlying neural mechanisms are not well understood, we hypothesize that each model requires specific features/classifiers to predict postoperative outcome in subtypes, given that each subtype has unique brain injure and clinical characteristics. This is also supported by previous studies suggesting that TLE patients with certain brain characteristics[34] or clinical features[33] may benefit from temporal surgery. Although there is debate about prognostic factors for surgical outcome in TLE, the presence of hippocampus sclerosis[34,35], a history of febrile seizure[36] and a low seizure frequency[33] were almost consistently reported to be associated with better outcomes. This perspective on building a stratified prediction model may be able to reveal underlying disease heterogeneity in surgery prognosis and guide a more individualized treatment in clinical practice.

A recent research has utilized the SuStaIn algorithm to explore the progression of epilepsy-related brain atrophy[37]. The study identified different subtypes of progression, including a cortical progression subtype and a non-cortical basal ganglia subtype in both focal and idiopathic generalized epilepsies. Additionally, a third hippocampus-driven progression subtype was specifically found in focal epilepsies. This subtype involved initial volume loss in the hippocampus, followed by the thalamus, and finally affecting other cortical areas. The observed spatiotemporal trajectory in the study aligns well with the current data. A recent work[14] also confirmed a similar sequence of regional changes in people with mesial temporal lobe epilepsy and hippocampal sclerosis, through an event-based disease progression modeling[38]. These suggest that the hippocampal-dominated trajectory may be one of the most significant features in TLE people.

This study had several limitations. Firstly, although the SuStaIn algorithm provided estimates of pathophysiological trajectories using cross-sectional MRI data, it is important to validate these findings with longitudinal data to confirm the disease progressions over time. MRI data at onset of epilepsy are needed to identify spatiotemporal patterns of brain atrophy to examine whether the spatiotemporal patterns of brain atrophy are caused by the progression of the disease or if they are the result of the initial brain injury. Secondly, this study benefited from image consistency, including the use of the same scanner, acquisition protocols, and image processing pipeline. Additionally, long-term follow-up data were available, including post-medication or post-operative clinical assessments. These factors strengthened the study's findings. However, to further validate the brain progression of each subtype, it would be advantageous to have larger samples with longitudinal data. There was a bit underpowered as the ratio of sample to feature is low. But we also verified the consistency of subtype at a relatively lower but acceptable spatial resolution. Thirdly, we described four distinct TLE biotypes using SuStaIn; they exhibited different in neuroanatomical signature, clinical phenotype and treatment outcome. However, elucidating potential mechanisms of subtypes is still challenging; future work is needed to contextualize the proposed biotypes of TLE with brain connectivity, cytoarchitecture[39], metabolism[40], neurotransmitter receptors and transporters[41], gene expression[42] and cognition-related brain function[43]. In addition, the current sample size is not enough to characterize a trajectory showing how treatment response changes as atrophy stage increases. Lastly, while a four-cluster solution was optimal for capturing temporal and phenotypic heterogeneity in our data, it is possible that more subtle and distinct subtypes may exist and warrant further investigation.

In conclusion, our study reveals three distinct pathophysiological trajectories of brain atrophy in TLE and identifies four subtypes with distinct neuroanatomical signatures. These subtypes exhibit diverse clinical characteristics and long-term antiepileptic outcomes, highlighting the heterogeneity of the disease and its implications for surgery prognosis. This imaging-based taxonomy provides valuable insights into the underlying biology of TLE and has important implications for personalized treatment approaches and prognostic assessment.

## Methods
### Participants

The primary sample consisted of 296 individuals with TLE (139 females, age = 27.2 ± 8.7 years) and 81 healthy subjects (39 females, age = 26.4 ± 6.7 years), recruited from January 2014 to August 2022 at West China Hospital. The inclusion criteria included that 1) patients were diagnosed with TLE according to the ILAE criteria[44]; 2) normal MRI or with unilateral hippocampal sclerosis (HS) evidence in keeping with electroencephalo-graph (EEG) findings; 3) no evidence of bilateral HS or of a secondary extrahippocampal lesion that may contribute to seizures. The exclusion criteria were as follows: 1) patients with other neurological disorder, psychiatric disorder or serious systemic disease; 2) with alcohol or other substances abuse; 3) with other structural lesions except HS according to ILAE classification[31] confirmed by postoperative histopathological examination. Patients underwent comprehensive multidisciplinary evaluations, combining the ictal semiology, ictal and interictal EEG, MRI and PET/CT if available, to localize the seizure focus. In addition, a validation sample consisted of 109 patients (61 females, age = 33.1 ± 10.4 years) diagnosed with TLE from three local hospitals (First affiliated hospital of Zhejiang University, N = 73; Fourth affiliated hospital of Zhejiang University, N = 21; First affiliated hospital of Hainan Medical University, N = 15).

After the initial assessments, patients were followed up every three months until April 2023 to determine their treatment options and outcomes. Based on the treatment option at the last follow-up, patients were divided into two groups: the operative group (OG) and the nonoperative medication group (MG). The treatment option was determined based on medical advice and patient preferences. The OG consisted of 152 patients who underwent anterior temporal lobe surgery. Patients who remained seizure-free after surgery were considered to have achieved an effective outcome, following the ILAE classification[44]. The MG included 144 patients who were treated with medication alone. An effective outcome in the medication group was defined as freedom from seizures for a duration of at least three times the longest interseizure interval before treatment or 12 months (whichever is longer), according to the criteria proposed by Kwan et al. (2011)[45]. We used the naturalistic data collected during routine clinical care; this is not a report of a randomized trial. This study was registered in the Chinese Clinical Trials Registry (number: ChiCTR2200062562) (https://www.chictr.org.cn/showproj.html?proj=176800). The data of registration was August 2022. Participants received travel compensation and remuneration. This study was approved by the local ethics committee of West China Hospital (ethics number: 2022-906) and informed consent was obtained from participants or their legal guardians.

## Image acquisition

High-resolution T1-weighted images were acquired on a 3 T MRI system (Trio; Siemens) with an 8-channel head coil at West China Hospital. Images were obtained in sagittal orientation using a spoiled gradient-recalled sequence with the main parameters: repetition time = 1900 ms; echo time = 2.26 ms; flip angle = 9°; slice thickness = 1 mm; field of view = 256 × 256 mm$^2$; voxel size = 1.0 × 1.0 × 1.0 mm$^3$.

## Image processing

T1-weighted images were processed using FreeSurfer (version 6.0, http://surfer.nmr.mgh. harvard.edu/). After visual inspections of segmentations, gray matter volumetric (GMV) measures values were estimated for 12 subcortical regions of interest (ROIs) including bilateral hippocampus, amygdala, caudate, nucleus accumbens, pallidum, putamen and thalamus. Cortical thickness (CT) measures were estimated for 64 cortical ROIs based on the DK atlas[46]. The ROI-wise GMV or CT measurements were first adjusted by regressing out the effects of sex, age, the square of age and total intracranial volume (TIV) using a regression model. Subsequently, the adjusted values were transformed as a z-score (i.e., normative deviations) relative to the healthy control group. Finally, we multiplied these z-scores by −1 so that the z-score increases as regional thickness/volume decreases.

## Subtype and Stage Inference (SuStaIn)

We employed an AI approach (i.e., SuStaIn)[18] to identify distinct patterns of spatiotemporal progression of brain atrophy from cross-sectional only MRI data and cluster individuals into groups (subtypes). Previous works has demonstrated ability of SuStaIn to identify diverse neurophysiological trajectories for brain disorders including fronto-temporal dementia, Alzheimer's disease and schizophrenia[18-20]. The methodology of SuStaIn has been presented previously[18]; we briefly describe the major parameter choices specific to the current study.

SuStaIn modeling needs an M x N z-score matrix as input. M is the number of cases (M = 296). N is the number of ROIs (N = 23). Due to computational complexity, SuStaIn algorithm typically applied no more than 25 ROIs for modeling in previous literatures[18,19]. Here, we selected a total of 23 gray matter ROIs (Supplementary Table 8) that reported reduced thickness/volume in mesial TLE patients relative to controls based on a recent ENIGMA-epilepsy structural MRI study[47]. We used the z-score thresholds (z = 1, 2, 3) as "waypoints" in the SuStaIn model[18]. We then ran the SuStaIn algorithm with 25 start points and 1,000,000 Markov Chain Monte Carlo (MCMC) iterations[18] to estimate the most likely sequence with spatiotemporal atrophy patterns (i.e., 'trajectory').

The model was fitted separately for k = 2-6 clusters ('trajectories')[18-20]. The optimal number of clusters with distinct trajectories was determined using the cross-validation information criterion (CVIC) and out-of-sample log-likelihood[18]. Lower value of CVIC represents better model fit. Supplementary Fig. 7a showed lowest CVIC when k = 3, indicating three distinct patterns of spatiotemporal progression of brain atrophy in MTLE. Similarly, log-likelihood increased indicating better model fit up until k = 3, after which no improvement was seen (Supplementary Fig. 7b). The 3-cluster model of SuStaIn was fitted to the whole sample. Final, the most probable sequence (i.e., the order of ROIs) at the population-level was evaluated for each 'trajectory'. The cumulative probability for each ROI to reach a particular z-score over SuStaIn stage is visualized using a positional variance diagram (Supplementary Fig. 8). For each individual, SuStaIn calculated the probability (ranging from 0 to 1) of belonging to which 'trajectory', and further assigned the individual into a sub-stage of the maximum likelihood 'trajectory' through MCMC iterations. The probability of maximum likelihood 'trajectory' is high across almost all SuStaIn stages (Supplementary Fig. 9). Note that SuStaIn assigned individuals who do not deviant obvious reduction in any ROI (here z scores of all ROIs <1) into the 'stage = 0'[18,19].

## Visualization of spatiotemporal trajectories of brain atrophy

To visualize the spatiotemporal patterns of pathophysiological progression across SuStaIn stages, we calculated the mean z-score of GMV across individuals belonging to the same stage of each SuStaIn 'trajectory'. ROI-wise GMV z-scores were mapped to a glass brain template using visualization tools implemented in ENIGMA Toolbox (https://enigma-toolbox.readthedocs.io/en/latest/index.html). To examine whether the SuStaIn stage (a continuous indicator derived from SuStaIn model) associate with neurophysiological and pathological process, we conducted the Spearman correlation between the SuStaIn stages and the degree of brain atrophy (i.e., the z scores of cortical thickness/subcortical volume). To characterize subtype-specific signature in neuroanatomical pathology, we conducted ROI-wise z score comparisons between any two subtype groups in addition to subtypes and healthy control group using two sample t-test. Multiple comparisons were corrected by FDR.

## Reproducibility of SuStaIn subtypes in another independent sample

We further examined the reproducibility of the SuStaIn trajectories in another independent validation sample including 109 patients diagnosed with temporal lobe epilepsy. Following the same image processing described in Methods 4.3, we extracted ROI-wise z-score for each patient. Subsequently, the SuStaIn trajectory was re-estimated based on the validation data using the same SuStaIn parameters with the modeling of discovery database (described in Methods 4.4). The spatiotemporal trajectory can be mathematically characterized as a sequence of ranked biomarkers (here n = 23). In addition, SuStaIn assigned each patient into a subtype, which allowed us to calculate average of z score map across individuals within the same subtype as a representation of subtype-specific atrophy signature. Pearson correlation coefficient was used as a quantitative coefficient to evaluate the consistency of z score map between discovery dataset and validation dataset. Spatial auto-correlation in brain map was corrected by a spatial autocorrelation-preserving permutation test (termed 'spin test')[48].

## Comparisons of clinical profiles between subtypes

Demographic, clinical and brain variables available for our cohort are described in the Table 1. These variables were statistically compared between subtypes, which involved two steps: (1) one-versus-all comparison. A one-versus-all approach was employed to compare each subtype to all individuals of other three subtypes to determine the subtype-specific characteristics, and (2) one-versus-one comparison. Each subtype was compared directly to each other subtype. The statistical comparisons were conducted using ANOVA with post-hoc Least Significant Difference (LSD) tests for continuous variables (age, age of onset, illness duration and TIV) or using Pearson's Chi-square test for categorical variables. Multiple comparisons were considered with FDR correction.

## Treatment outcomes in subtypes

In this exploratory analysis, we explored whether neuroanatomical subtype classification based on baseline MRI will relate to differential treatment outcomes to medications or anterior temporal lobe operative treatment. In the medications group (MG) including 144 patients with TLE who received medications, 21 patients reported seizure-freedom at the follow-up (mean interval is 56.3 months). In the operative group (OG) including 152 TLE patients before taking anterior temporal lobe operative treatment, 96 individuals following operative treatment reported seizure-freedom at follow-up (mean interval is 64.1 months). Using the baseline MRI data, the SuStaIn model assigned each individual into one of four subtypes. We compared the difference of the follow-up seizure-freedom rate among the four subtypes using Pearson's Chi-square test.

**Predicting prognosis of surgery by SuStaIn subtype-based prediction model**

To examine whether the SuStaIn subtype information at baseline could help to predict the prognosis of surgery at follow-up for a given patient, we conducted machine learning procedures to predict treatment outcome in a sub-sample of 145 post-surgery follow-up subjects. Here, we described how to train a support vector machine (SVM) classifier (Supplementary Fig. 10). Specifically, we applied ten-fold cross-validation to obtain train data and test data. In each fold, 90% of subjects was used as a training set, and the left-out 10% subjects were used as a test set. In training set, the classifier features included the baseline clinical variables, MRI variables, or both. Principal component analysis (PCA) was used to reduce feature dimension. The first N principal components (PCs), which explained beyond 95% of the variance of all features, were used to train a SVM classifier to classify the subject who achieves seizure freedom (OG+) or not (OG-) after surgery. Three commonly used SVM kernel functions (linear, RBF and polynomial) were used. The test set patient's outcome was predicted based on the built SVM classifier. Prediction performance was measured by sensitivity, specificity and accuracy. We also calculated Youden Index (sensitivity+specificity-1) as a comprehensive assessment of both sensitivity and specificity. To further examine whether the prediction performance is significantly better than random predictions, we used a permutation test to evaluate significance by random permutation of predictive label (Supplementary Materials).

## Exploratory analysis

To evaluate the stability of SuStaIn at a relative lower spatial resolution, the 23 ROI features were down sampled to 13 features by merging regions within the same cortical lobe. We investigated the difference of total intracranial volume (TIV) between subtypes and healthy controls; we also examine whether TIV was associated with specific clinical features (Supplementary Materials).

## Reporting summary

Further information on research design is available in the Nature Portfolio Reporting Summary linked to this article.

## Data availability

The raw image and clinical data are protected and are not available due to data privacy laws. Requests for raw data can be made to the corresponding author and will be promptly reviewed by the local ethics committee to verify whether the request is subject to any intellectual property or confidentiality obligations. The processed data and statistical results are provided in the Supplementary Information/Source Data file. Source data are provided with this paper.

## Code availability

T1-weighted images were processed using FreeSurfer (version 6.0, http://surfer.nmr.mgh.harvard.edu/). Raw code of the SuStaIn algorithm are available on the UCL-POND GitHub (https://github.com/ucl-pond). The visualization of ROI-wise z-score was conducted using ENIGMA Toolbox (https://enigma-toolbox.readthedocs.io/en/latest/index.html). Statistical analyses, including correlation analysis, t-test, ANOVA etc., were conducted using MATLAB (version: R2018b) and SPSS Statistics (version: 26.0).

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

## Acknowledgements

This work was supported by the grant from Science and Technology Innovation 2030-Brain Science and Brain-Inspired Intelligence Project (No. 2022ZD0212800 to Y.J.). This work was supported by National Natural Science Foundation of China (No. 82202242 to Y.J., No. 82171443 to D.A., No. U21A20393 to D.Z.). This work was supported by the projects from China Postdoctoral Science Foundation (No. BX2021078 and 2021M700852 to Y.J.). This work was supported by the Shanghai Sailing Program (22YF1402800 to Y.J.) from Shanghai Science and Technology Committee. This work was supported by Natural Science Foundation of Sichuan Province (No. 2022NSFSC1483 to W.L.). This work was supported by Chengdu Science and technology Bureau Program (No. 2019-YF09-00215-SN to D.Z.). The funders had no role in study design, data collection and analysis, decision to publish or preparation of the manuscript.

## Author contributions

Y.J. and D.A. led the project. Y.J., W.L. and D.A. were responsible for the study concept and the design of the study. C.L., D.Y. and D.Z. made substantial contributions to the manuscript and provided crucial advice for the study. Y.J. and W.L. analyzed the data, created the figures and wrote the original manuscript. J.L., X.L., H.Z., X.S., L.L., K.W., Q.L., J.F. and Q.G. contributed to the data acquisition. L.J. helped with the data analysis and commented on the manuscript.

## Competing interests

The author declares no competing interests.

## Additional information

[1]Institute of Science and Technology for Brain-Inspired Intelligence, Fudan University, Shanghai, China. [2]Department of Neurology, West China Hospital, Sichuan University, Chengdu, Sichuan, China. [3]Department of Geriatrics, West China Hospital, Sichuan University, China National Clinical Research Center for Geriatric Medicine, Chengdu, China. [4]Huaxi MR Research Center, Department of Radiology, West China Hospital, Sichuan University, Chengdu, Sichuan, China. [5]Department of Neurosurgery, West China Hospital, Sichuan University, Chengdu, Sichuan, China. [6]Epilepsy Center, Department of Neurology, The first affiliated hospital, School of Medicine, Zhejiang University, Hangzhou, Zhejiang, China. [7]Department of Neurology, The first affiliated hospital, Hainan Medical University and the Key Laboratory of Brain Science Research and Transformation in Tropical Environment of Hainan Province, Haikou, Hainan, China. [8]Department of Neurology, The fourth affiliated hospital, School of Medicine, Zhejiang University, Yiwu, Zhejiang, China. [9]Psychological Medicine Center, The First Affiliated Hospital of Xinjiang Medical University, Urumqi, Xinjiang, China. [10]The Clinical Hospital of Chengdu Brain Science Institute, MOE Key Lab for Neuroinformation, School of Life Science and technology, University of Electronic Science and Technology of China, Chengdu, China. [11]High-Field Magnetic Resonance Brain Imaging Key Laboratory of Sichuan Province, Center for Information in Medicine, University of Electronic Science and Technology of China, Chengdu, China. [12]Research Unit of NeuroInformation (2019RU035), Chinese Academy of Medical Sciences, Chengdu, China. [13]These authors contributed equally: Yuchao Jiang, Wei Li. ✉e-mail: yuchaojiang@fudan.edu.cn; zhoudong66@yahoo.de; chengluo@uestc.edu.cn; andongmei2010@gmail.com

