## [Peer Review File · Nature Communications]

Identification of four biotypes in temporal lobe epilepsy via machine learning on brain imagesREVIEWER COMMENTS

Reviewer #1 (Remarks to the Author):

Summary of the key results:

The authors intended to identify different patterns of gray matter loss in individuals with temporal lobe epilepsy (TLE) using a machine learning algorithm called Subtype and Stage Inference (SuStaln) on MRI data. They grouped patients into subtypes based on how and when the loss occurred and examined the differences in brain structure, clinical features, and treatment outcomes among these subtypes.

They identified three different patterns of atrophy, referred to as 'trajectory' 1, 'trajectory' 2, and 'trajectory' 3. In 'trajectory' 1, the initial decrease in volume was observed in the left hippocampus, followed by the left thalamus. It then progressed to the right thalamus before finally affecting the left entorhinal cortex and cerebral cortex. In 'trajectory' 2, volume loss began in the right hippocampus, followed by the right thalamus. It then spreads to the left thalamus and left hippocampus before impacting the cerebral cortex. 'Trajectory' 3 was characterized by a primary reduction in the cortex, specifically involving both the middle and superior frontal lobes on both sides. This atrophy later extended to the parietal, occipital, and temporal lobes. Following that, it affected subcortical areas such as the hippocampus and thalamus.

These trajectories were replicated in an independent validation cohort.

The SuStaln algorithm computed the likelihood of each patient being a part of a particular 'trajectory' and then assigned them to a sub-level within that trajectory.

When comparing ROI-wise z scores (degree of thickness/volume decrease) between each subtype and the healthy control group, the researchers identified four distinct neuroanatomical signatures. These signatures were labeled as follows: 1. The left hippocampus-predominant signature (subtype1) 2. The right hippocampus-predominant signature (subtype2) 3. The cortex-predominant signature (subtype3) 4. The 'normal' signature (subtype4) Subtypes 1 and 2 exhibited the most severe atrophy in the ipsilateral hippocampus. Subtype 3 showed gray matter loss primarily in the neocortices, while subtype 4 demonstrated increased gray matter volume, with the most significant

enlargement observed in the amygdala. The researchers also noted significant differences in various clinical variables among the four subtypes. These variables included age of onset, illness duration, seizure lateralization, MRI hippocampal sclerosis (HS) rate, history of febrile seizures, aura, and treatment outcomes.

Originality and significance:

The paper utilizes a strong data-driven method. However, it is not clear whether these subtypes are genuinely new or simply a fresh designation for TLE with left HS, TLE with right HS, TLE with negative MRI, and TLE with enlarged amygdala. A significant amount of existing literature confirms different characteristics within these four groups of TLE. This aspect requires further discussion and elaboration (see below).

Data & methodology: validity of the approach, quality of data, quality of presentation:

Data and methods are sound and presented clearly.

Appropriate use of statistics: Yes.

Suggested improvements: experiments, data for possible revision:

Based on the comments above, the subtypes they described were mostly connected to the presence and location of MRI findings of hippocampal sclerosis (HS), which is not surprising. It is also known that the side of seizure lateralization corresponds to the side with HS and that patients with MRI evidence of HS generally have a worse response to medications and experience seizures at a younger age compared to those with normal MRI results upon visual examination. There are numerous publications that demonstrate these distinct characteristics in patients with HS, particularly worse findings on the left side. Additionally, there are differences in the response to surgical treatment, with those without hippocampal atrophy usually experiencing worse outcomes. These points need further discussion.

Another point to be addressed is whether the "spatiotemporal patterns of brain atrophy" are caused by the progression of the disease or if they are the result of the initial brain

injury (in other words, could they have been present since the onset of epilepsy?). How can the authors differentiate between these possibilities in a cross-sectional group of individuals?

In line 373, the authors mentioned that they multiplied z-scores by -1 so that the z-score increases as regional thickness/volume decreases. This is a bit confusing when looking at Figures, for example, Figure S3: "ROI-wise z score of bilateral amygdala volume. Note that a negative z-score represents a larger gray matter volume relative to the healthy control group." Is it a larger volume or more atrophy? Please clarify.

Reviewer #2 (Remarks to the Author):

Jiang, Li, and coauthors employ the Subtype and Stage Inference (SuStaln) algorithm to discover subtypes of temporal lobe epilepsy (TLE) based on patterns of brain atrophy (or lack thereof). The authors find three different atrophy patterns among TLE patients, plus a subgroup with no atrophy. Interestingly, the no-atrophy subgroup showed greater amygdala volume compared to controls and showed better 5-year treatment response to medication (as opposed to surgery), while the three atrophy subtypes showed better response to surgery.

The findings in this study are exciting examples of how data-driven approaches can identify measurable biological features that help distinguish subgroups of individuals that vary along clinically-relevant dimensions. The 5-year follow-up data in this study, validation sample, and relatively large sample size are great strengths of this study. However, there are flaws in the use and interpretation of SuStaln, limiting the interpretability of certain findings, while other potentially interesting findings are underdeveloped. Details follow.

Note that I am an expert in the SuStaln algorithm, but am not an expert in TLE.

METHODOLOGICAL CONCERNS

1) The SuStaln input information is not adequately described. The authors should disclose

what z-score thresholds are used as “waypoints” in the SuStaln model (Young 2018 Nat Comms). It is also not possible to infer this from the figures. The z-score range 0-2 in Figure 1 but 0-3 in Figure 2. Why this discrepancy?

2) The study is a bit underpowered as the feature to sample ratio is quite low. Assuming $z=1,2,3$ (but I'm guessing here, see above), that's 3 temporal features and 25 regional features, totaling 75 features. With $n=296$ subjects, that's slightly less than 4 subjects per feature, which is not ideal. The authors should ask themselves if all 25 regional features are necessary. Do they all differ between controls and patients (if not their inclusion is not necessary)? Is it possible to combine features or reduce dimensionality in some way? If not, the authors should at least acknowledge being somewhat underpowered in the limitations.

3) One aspect of the study that breeds optimism re: point 2 above is the validation analysis in the separate sample. However, reporting of this information is quite sparse. There is no mention of any validation procedures in the Methods (outside of describing the validation sample), and there is only one supplemental figure with hardly any mention. A quantitative comparison between the discovery and validation sets is warranted. Could the authors correlate the (subtype vs control) effect maps? And/or correlate the event position of features between the two datasets?

4) Some aspects of the subtypes could use to be described in more detail:

a) Please include a (preferably cross-validated positional variance diagram so the reader can get a sense of the confidence in each subtype sequences.

b) It would be helpful to describe comparisons of subtypes between one another in addition to between subtypes and controls.

c) What stages are being visualized in Figure 1A-C?

d) It would help to include a supplemental figure plotting subtype probability across stages, stratified by subtypes. This helps visualize whether there are “cross-over events” — moments in the subtype sequence when two subtypes “merge” (see e.g. Aksman et al., 2023 Brain; Young, Vogel et al., 2023 Brain). These are important to identify because, if present, they make interpretation somewhat more challenging.

5) The authors seem to misunderstand the meaning of Stage 0 individuals. If an individual is stage 0, they do not have enough abnormal feature values to be classified into a subtype. However, SuStaln will assign them subtypes and probabilities anyway. These probabilities are likely meaningless as they are not based on meaningful variation in feature values. Therefore, any analyses involving SuStaln stage or subtype probabilities should exclude Stage 0 (no-atrophy) individuals. This includes e.g. Fig 1 e.g., the logistic regression analysis, and the main findings of Table 1 — how can one covary for stage here when one of the “subtypes” always has a stage of 0?

6) There are several issues with the “sp-score” analysis, as follows:

a) It is important to note this is not a normally distributed variable (not even close), which is a consideration in the use of linear models.

b) My understanding is that the “sp-score” is just a rebranding of the maximum-likelihood subtype probability. If so, the authors may misunderstand what this variable is. The LR analysis is essentially just predicting how stably a subject belongs to *their mostly likely* subtype. This analysis can be thought of as establishing which non-MR variables could contribute to subtype confidence. But this analysis would also change depending on which subtype probability is being used — when using the maximum-likelihood probability, different information sources are being mixed together. It would make more sense to perform the LR on each subtype probability separately.

c) Related: On multiple occasions, the author claim the sp-score is “interpreted” as something, based on the LR analysis. That doesn’t make sense. The sp-score already has a very clear interpretation based on the SuStaln model formulation, and it is entirely to do with how much an individual’s MR pattern matches that of their maximum likelihood canonical subtype. So, in this analysis, the authors are merely describing things associated with subtype probability

d) How were the features for the sp-score analysis chosen? Why isn’t SuStaln stage included?

e) The logistic regression approach used is likely optimistically biased, and does not qualify as “AI”. Was there any cross-validation? Parameter tuning? Was a linear kernel used? The details are sparse but, without them, the reader assumes this model is overfit.

f) Finally, this analysis is generally oversold and sensationalized. On line 282 the authors

claim to have “introduced” this metric, when it is a main output of the SuStaln algorithm and is commonly used across all SuStaln papers, either as a means of exclusion (e.g. Collij et al., 2022 Neurology) or to compare subtype confidence with other measures (e.g. Vogel et al., 2021 Nat Med, Young et al., 2023 Brain).

ANALYTIC / CONCEPTUAL CONCERNS

7) I was surprised and fascinated to see a subtype effect on total intracranial volume — individuals with the cortical subtype apparently had significantly larger crania than the other three subtypes. Given important recent associations between head size and disease (e.g. Seidlitz et al., 2022 bioRxiv; Adams et al., 2016 Nat Neuro), it is of great interest to establish whether this finding is biological or methodological. Was TIV different between that subtype and controls? Can the authors plot this particular association? Also, if there is a subtype difference in TIV, does that complicate the fact that authors must control volumetric effects by TIV? The TIV finding is not mentioned at all in the results or discussion, but I believe it deserves attention.

8) Are any of the main variables in Table 1 associated with SuStaln stage? Were there any Subtype x Stage interactions?

9) The most important finding of the study is the subtype effect on treatment response. However, the effect is not so much a subtype effect as an atrophy vs. no atrophy effect. Those without measurable brain atrophy are more likely to respond to medication and those with brain atrophy more likely respond to operation. To establish whether this is truly a subtype effect or a brain atrophy effect, it would be helpful for the authors to plot likelihood of treatment response *across* stage *within* subtype. This can be done with a sliding window approach. Is it the case that the more atrophy a subject has, the more likely they respond to operation (and/or don't respond to medication)? Or is purely a subtype effect where the results don't vary across stage, only across atrophy vs no atrophy subtypes?

10) Treatment response is an extremely clinically relevant outcome measure. The authors have shown subtype effects that are very interesting, but it would be helpful to better understand the degree to which this information aids prediction of such outcomes. Rather than the sp-score analysis (which I'm not sure adds much to the study), could the authors perform an LR analysis predicting treatment outcome? I would be interested to see either an inferential analysis where the independent contribution of subtype is quantified compared to other variables, or a machine-learning analysis with and without subtype included as a feature to measure its contribution to treatment-outcome prediction.

MINOR ISSUES

* The authors first regress out covariates and then z-score data. Typically, one would use a "w-score" technique (La Joie et al. 2012 J Neurosci), where covariates are modeled in controls and then applied to all subjects during the z-scoring procedure. This is the approach used in Young et al. 2018 and most subsequent SuStaln papers.

Line 427 — "The statistical comparisons were conducted using ANOVAs with appropriate post-hoc tests for continuous variables..." — Could the authors be more specific?

* TIV is not necessary as a covariate for cortical thickness (Schwarz et al. 2016 Neuroimage: Clin)

* The authors make a big deal out of the finding of Stage 0 individuals having larger amygdalae compared to controls. But stats on the amygdala (nor hippocampus) are not shown in Figure 3.

Manuscript Title: Identification of four biotypes in temporal lobe epilepsy via brain imaging-based machine learning

Yuchao Jiang, Wei Li, et al: Response to Reviewers

Reviewer #1 (Remarks to the Author):

Summary of the key results:

The authors intended to identify different patterns of gray matter loss in individuals with temporal lobe epilepsy (TLE) using a machine learning algorithm called Subtype and Stage Inference (SuStaln) on MRI data. They grouped patients into subtypes based on how and when the loss occurred and examined the differences in brain structure, clinical features, and treatment outcomes among these subtypes.

They identified three different patterns of atrophy, referred to as 'trajectory' 1, 'trajectory' 2, and 'trajectory' 3. In 'trajectory' 1, the initial decrease in volume was observed in the left hippocampus, followed by the left thalamus. It then progressed to the right thalamus before finally affecting the left entorhinal cortex and cerebral cortex. In 'trajectory' 2, volume loss began in the right hippocampus, followed by the right thalamus. It then spreads to the left thalamus and left hippocampus before impacting the cerebral cortex. 'Trajectory' 3 was characterized by a primary reduction in the cortex, specifically involving both the middle and superior frontal lobes on both sides. This atrophy later extended to the parietal, occipital, and temporal lobes. Following that, it affected subcortical areas such as the hippocampus and thalamus.

These trajectories were replicated in an independent validation cohort.

The SuStaln algorithm computed the likelihood of each patient being a part of a particular 'trajectory' and then assigned them to a sub-level within that trajectory.

When comparing ROI-wise z scores (degree of thickness/volume decrease) between each subtype and the healthy control group, the researchers identified four distinct neuroanatomical signatures. These signatures were labeled as follows: 1. The left hippocampus-predominant signature (subtype1) 2. The right hippocampus-predominant signature (subtype2) 3. The cortex-predominant signature (subtype3) 4. The 'normal' signature (subtype4) Subtypes 1 and 2 exhibited the most severe atrophy in the ipsilateral hippocampus. Subtype 3 showed gray matter loss primarily in the neocortices, while subtype 4 demonstrated increased gray matter volume, with the most significant enlargement observed in the amygdala. The researchers also noted significant differences in various clinical variables among the four subtypes. These variables included age of onset, illness duration, seizure lateralization, MRI hippocampal sclerosis (HS) rate, history of febrile seizures, aura, and treatment outcomes.

Originality and significance: The paper utilizes a strong data-driven method. However, it is not clear whether these subtypes are genuinely new or simply a fresh designation for TLE with left HS, TLE with right HS, TLE with negative MRI, and TLE with enlarged amygdala. A significant amount of existing literature confirms different characteristics within these four groups of TLE. This aspect requires further discussion and elaboration (see below).

Data & methodology: validity of the approach, quality of data, quality of presentation:

Data and methods are sound and presented clearly.

Appropriate use of statistics: Yes.

Response: We thank the reviewer for the positive appraisal of our work. We are grateful for the detailed feedback focused on a few key concerns. As anticipated by the reviewer, addressing these issues rigorously and comprehensively has entailed major additional analyses, which have now been included in the paper as described in more detail below.

[BLACK] - ORIGINAL COMMENT

[BLUE] - RESPONSE TO COMMENT

[HIGHLIGHTED] - NEW TEXT AND FIGURE/TABLE CHANGES

Ref 1/1

Suggested improvements: experiments, data for possible revision:

Based on the comments above, the subtypes they described were mostly connected to the presence and location of MRI findings of hippocampal sclerosis (HS), which is not surprising. It is also known that the side of seizure lateralization corresponds to the side with HS and that patients with MRI evidence of HS generally have a worse response to medications and experience seizures at a younger age compared to those with normal MRI results upon visual examination. There are numerous publications that demonstrate these distinct characteristics in patients with HS, particularly worse findings on the left side. Additionally, there are differences in the response to surgical treatment, with those without hippocampal atrophy usually experiencing worse outcomes. These points need further discussion.

Response: Thanks for the suggestion. This is indeed a very important point. These are highly relevant for TLE and clinical characterization. We think it necessary to discuss the SuStaln subtypes with the four known anatomical types (TLE with left HS, TLE with right HS, TLE with negative MRI, and TLE with enlarged amygdala).

1. In our data, we did observe that patients with HS+ show younger age of onset compared to those HS- patients ($t=-3.49$, $p=0.001$). In addition, we found that patients with HS- experience worse surgical outcomes compared to those HS+ patients ($\chi^2=5.99$, $p=0.014$) (**Supplementary Table S5.1**). These findings are consistent with previous studies (Malmgren, et al., 2012; Blumcke, et al., 2013; Jehi et al., 2015; Lamberink, et al., 2020). To examine whether the clinical differences among SuStaln subtypes are affected by HS, we re-analyzed the correlations between clinical features and subtype with HS effect as a covariate using a linear regress or logistic regress model ($Y \sim X + C + \epsilon$). Here, clinical variable is dependent variable (Y); SuStaln subtype is independent variable (X); and HS is covariate variable (C). After controlling HS effect, we still found significant associations of SuStaln subtype with age of onset, illness duration and medication outcomes (**Supplementary Table S5.2**). This suggests that the clinical differences between SuStaln subtypes remain significant after controlling HS effect. We added this part of the supplementary analysis to the Supplementary

material and revised the main text (provided at the end).

2. We also acknowledge that the neuro-structural features of SuStaln subtypes were mostly associated with imaging characteristics of presence and location of MRI HS within four groups. This again confirms the brain structural heterogeneity of TLE and highlights the necessary to a reliable imaging-based taxonomy to identify a more homogeneous sub-population of individuals with shared neurobiological attributes. In addition, this work expanded distinct spatiotemporal trajectories of neuro-structural subtypes using a validated disease progression modeling method, whose stability is also confirmed with longitudinal data. We are also very willing to add discussion on this part, as suggested by the reviewer (also provided at the end).
3. SuStaln is a data-driven classification algorithm, which relies only on T1-weighted brain image to define subtypes without any prior clinical information (of course, TLE diagnosis is still necessary). In clinical practice, the criteria of HS or seizure lateralization relies on comprehensive evaluations (combined with other auxiliary examinations), by experienced clinicians. Here, both of data-driven result (SuStaln subtype) and clinical evidence reveal a high brain structural heterogeneity within individuals diagnosed with TLE. But we hold that SuStaln subtype is not simply a fresh designation for known anatomical types (left HS, right HS, negative MRI, and enlarged amygdala). This is supported by the proportion of MRI HS+ for SuStaln subtype 3 and subtype 4 (about 60% HS- patients); the proportion indicates that the two subtypes include ~40% individuals with MRI negative type.
4. In only individuals with MRI HS+, we still observed differences of neuro-structural features between SuStaln subtypes (**please see following Table R1**), suggesting that it still exhibits highly individual variability even in a relatively homogeneous clinical subgroup with well-defined brain injury (e.g., MRI HS+). It also suggests that the SuStaln subtype is not simply a fresh designation for existing HS+ or HS- subgroup.

Table R1. Comparison among four SuStaln subtypes only in individuals with MRI HS+.

Region	F	P
Left-Thalamus	17.9	<0.001
Left-Putamen	1.9	0.131
Left-Hippocampus	111.5	<0.001
Right-Thalamus	23.7	<0.001
Right-Pallidum	4.1	0.008
Right-Hippocampus	248.9	<0.001
Right_caudalmiddlefrontal	16.8	<0.001
Right_paracentral	18.5	<0.001
Right_parsopercularis	11.9	<0.001
Right_parstriangularis	14.4	<0.001
Right_precentral	14.0	<0.001
Right_precuneus	19.5	<0.001
Right_superiorfrontal	17.8	<0.001
Left_caudalmiddlefrontal	10.0	<0.001

Left_entorhinal	2.5	0.063
Left_fusiform	8.7	<0.001
Left parahippocampal	3.9	0.01
Left_paracentral	8.3	<0.001
Left_precentral	13.4	<0.001
Left_precuneus	10.6	<0.001
Left_superiorfrontal	13.5	<0.001
Left_temporalpole	3.6	0.015
Left_transversetemporal	12.5	<0.001

Reference

Malmgren, K. and M. Thom, *Hippocampal sclerosis--origins and imaging. Epilepsia*, 2012. 53 Suppl 4: p. 19-33.

Blumcke, I., et al., *International consensus classification of hippocampal sclerosis in temporal lobe epilepsy: a Task Force report from the ILAE Commission on Diagnostic Methods. Epilepsia*, 2013. 54(7): p. 1315-29.

Lamberink, H.J., et al., *Seizure outcome and use of antiepileptic drugs after epilepsy surgery according to histopathological diagnosis: a retrospective multicentre cohort study. Lancet Neurol*, 2020. 19(9): p. 748-757.

Jehi, L., et al., *Development and validation of nomograms to provide individualised predictions of seizure outcomes after epilepsy surgery: a retrospective analysis. Lancet Neurol*, 2015. 14(3): p. 283-90.

<<The following changes have been made to the Main Text>>

In **Results 2.1 Clinical characterization of subtypes**

We also observed that patients with MRI evidence of HS (HS+) show younger age of onset compared to those with normal MRI results (HS-) upon visual examination ($t=-3.49$, $p=0.001$). In addition, we found that patients with HS- experience worse surgical outcomes compared to those HS+ patients ($\chi^2=5.99$, $p=0.014$). To examine whether the clinical differences among SuStaln subtypes are affected by HS, we re-analyzed the correlations between clinical features and subtype with HS effect as a covariate (Supplementary Materials). We still found significant correlations of SuStaln subtype with age of onset ($t=-3.51$, $p=0.001$), illness duration ($t=-3.15$, $p=0.002$) and medication outcomes ($\chi^2=5.64$, $p=0.018$) after controlling HS effect (Supplementary Materials)

In **Discussion**

The SuStaln subtypes show similar clinical characteristics with four known TLE types (TLE with left HS, TLE with right HS, TLE with negative MRI, and TLE with enlarged amygdala) [27, 29]. The neuro-structural features of SuStaln subtypes were mostly associated with presence and location of MRI HS [6]. This again confirms the brain structural heterogeneity within individuals diagnosed with TLE. In clinical studies, TLE with HS generally have a worse response to medications and experience seizures at a younger age compared to those with normal MRI upon visual examination [30, 31]. This is also observed in SuStaln subtype 1, which includes most of patients with left side HS. Additionally, previous studies

reported that those individuals without hippocampal atrophy usually experience worse responses to surgical treatment [32, 33]. In our study, we also observed that patients with HS- experience worse surgical outcomes compared to those HS+ patients. We suspect that this may be one of the reasons for the poor surgical outcomes of SuStaln subtype 4. In short, the differences between data-driven subtypes are in part consistent with known clinical features.

<<The following changes have been made to the Supplementary Materials>>

In *Methods S5 Hippocampal Sclerosis Analysis*

In our data, we examined the associations between HS and other clinical variables (age of onset, illness duration, medication outcome and surgery outcome) using a linear regress or logistic model analysis. We observed that patients with HS+ show younger age of onset compared to those HS- patients ($t=-3.49$, $p=0.001$). In addition, we found that patients with HS- experience worse surgical outcomes compared to those HS+ patients ($\chi^2=5.99$, $p=0.014$) (**Supplementary Table S5.1**). To examine whether the clinical differences among SuStaln subtypes are affected by HS, we re-analyzed the correlations between clinical features and subtype with HS effect as a covariate using a linear regress or logistic regress model ($Y \sim X + C + \varepsilon$). Here, clinical variable is dependent variable (Y); SuStaln subtype is independent variable (X); and HS is covariate variable (C). After controlling HS effect, we still found significant associations of SuStaln subtype with age of onset, illness duration and medication outcomes (**Supplementary Table S5.2**). This suggests that the clinical differences between SuStaln subtypes remain significant after controlling HS effect.

Supplementary Table S5.1 Results of linear/logistic regress model with HS as an independent variable

Y variable	ANOVA & chi-square test		X variable (HS effect)		
	F & χ^2	p	β	t & Wals	p
Age of onset	12.20	0.001	-4.48	-3.49	0.001
Illness duration	3.76	0.054	2.24	1.94	0.054
Medication outcome	2.89	0.089	-0.85	2.93	0.087
Surgery outcome	5.99	0.014	-1.13	5.94	0.014

Supplementary Table S5.2 Results of linear/logistic regress model with SuStaln subtype as an independent variable

Y variable	ANOVA & chi-square test		X variable (SuStaln subtype effect)		
	F & χ^2	p	β	t & Wals	p
Age of onset	18.85	<0.001	5.17	4.34	<0.001
Age of onset #	12.51	<0.001	4.33	3.51	0.001
Illness duration	13.73	<0.001	-3.80	-3.71	<0.001
Illness duration #	6.88	0.001	-4.00	-3.15	0.002
Medication outcome	8.43	0.004	1.50	8.47	0.004
Medication outcome #	8.58	0.014	1.39	5.64	0.018
Surgery outcome	4.42	0.036	-1.03	4.42	0.036
Surgery outcome #	8.34	0.015	-0.79	2.37	0.123

represents a linear/logistic regress model with HS as an additional covariate variable.

Ref 1/2

Another point to be addressed is whether the "spatiotemporal patterns of brain atrophy" are caused by the progression of the disease or if they are the result of the initial brain injury (in other words, could they have been present since the onset of epilepsy?). How can the authors differentiate between these possibilities in a cross-sectional group of individuals?

Response: Thanks for the comment. This is indeed a very interesting point regarding whether the "spatiotemporal patterns of brain atrophy" are caused by the progression of the disease or if they are the result of the initial brain injury. To further clarify it, we took two exploratory analyses as follow.

1. We followed up brain MRI data of the part of samples (n=23, without surgery until follow-up). The average of interval time between the baseline scanning and follow-up scanning is 39.0 months (SD=16.8 months), range from 10.5 to 76.7 months. Using this subsample, we re-estimated the SuStaln subtype labels of these individuals using their follow-up MRI data. We would like to examine whether the subtype label at follow-up keeps consistent with the baseline label; if consistent, it suggests that 'spatiotemporal patterns of brain atrophy' (i.e., individual belong to which subtype) may not be affected by disease progress. In fact, we observed that subtype labels remained consistent for almost all patients at baseline and follow-up (following **Figure S6**), except for two patients with subtype 4 (i.e., stage=0) who had converted to subtype 1 and subtype 3, respectively. This result indicates that the initial representation belonging to which 'spatiotemporal patterns of brain atrophy' (i.e., SuStaln subtype) may not be affected by disease progress.

Figure S6. Subtype labels remain consistent for almost all patients at baseline and follow-up.

2. We divided all of individuals with TLE (n=296) into two disease subgroup according to their disease durations (median=9.5 years is set at cutoff threshold), yielding a short-

term subgroup (n=148, mean disease duration= 4.8 ± 2.7 years) and a long-term subgroup (n=148, mean disease duration= 17.5 ± 7.4 years). Such a subgrouping rule take into account the same size of subsamples. We re-estimated the 'spatiotemporal patterns of brain atrophy' (i.e., SuStaln trajectory) in each subgroup, separately. We found there was a similar pattern of the three trajectories (left hippocampus-led, right hippocampus-led and cortex-led) in the two disease subgroups (**Figure S7**). This result suggests again that the distinct spatiotemporal patterns of brain atrophy may not be affected by disease progress.

Taken together, the two exploratory analyses above suggest that the "spatiotemporal patterns of brain atrophy" may be a determinate evolution of brain pathophysiology since epilepsy early stage. In other words, since certain initial brain injury is established, it is less likely to shift from one trajectory pattern (i.e., subtype) to another. This assumption is also supported by **Figure S4**, which shows that the probability of maximum likelihood subtype is high across all SuStaln stage, indicating that there was no "cross-over events" in the subtype sequence.

Nevertheless, MRI data at onset of epilepsy are still needed to re-identify whether spatiotemporal patterns of brain atrophy present since the onset of epilepsy. We also emphasized this point in the limitation in the revised manuscript.

<<The following changes have been made to the Main Text>>

In Results 2.1 Distinct pathophysiological progressions of brain atrophy

We also estimated spatiotemporal trajectories of brain atrophy in a short-term subsample (n=148, mean disease duration= 4.8 ± 2.7 years) and a long-term subsample (n=148, mean disease duration= 17.5 ± 7.4 years), separately (Supplementary Materials). There was a similar pattern of the three trajectories in the two disease subsample (Supplementary Materials). This suggests that the distinct spatiotemporal patterns of brain atrophy may not be affected by disease progress.

In Results 2.3 Reproducibility of SuStaln subtypes

To examine whether the subtype label keeps consistency as disease progresses, we followed up brain MRI data of a subsample (n=23, average of interval time= 39.0 ± 16.8 months). The labels of subtype at follow-up remained consistent with baseline for almost all patients (Supplementary Materials), suggesting that the initial classification belonging to which SuStaln subtype may not be affected by disease progress.

In Discussion

MRI data at onset of epilepsy are needed to identify spatiotemporal patterns of brain atrophy to examine whether the spatiotemporal patterns of brain atrophy are caused by the progression of the disease or if they are the result of the initial brain injury.

<<The following changes have been made to the Supplementary Materials>>

In *Methods S3 MRI Follow-up Analysis*

We follow up brain MRI data of the part of individuals with TLE (n=23, without surgery until follow-up). The average of interval time between the baseline scanning and follow-up scanning is 39.0 months (SD=16.8 months), range from 10.5 to 76.7 months. Using this subsample, we re-estimated the SuStaln subtype labels of these individuals using their follow-up MRI data. We examined whether the subtype label at follow-up keeps consistent with the baseline label. We found that subtype labels remained consistent for almost all patients at baseline and follow-up (Supplementary Figure S6), except for two patients with subtype 4 (i.e., stage=0) who converted to subtype 1 and 3 at follow-up, respectively. This result suggests that once certain initial brain injury is established, it is less likely to shift from one trajectory pattern (i.e., subtype) to another. This assumption is also supported by Supplementary Figure S4, which shows that the probability of maximum likelihood subtype is high across all SuStaln stage, indicating that there was no “cross-over events” in the subtype sequence.

Figure S4. Probability of maximum likelihood trajectory is high across almost all SuStaln stages.

Figure S6. Subtype labels remain consistent for almost all patients at baseline and follow-up.

In *Methods S4 Disease Duration Subsample Analysis*

We divided all of individuals with TLE (n=296) into two disease subgroup according to their disease durations (cutoff = median value (i.e., 9.5 years)), yielding a short-term subgroup (n=148, mean disease duration=4.8 ± 2.7 years) and a long-term subgroup (n=148, mean disease duration=17.5 ± 7.4 years). Such a subgrouping rule take into account the same size of subsamples. We re-estimated the 'spatiotemporal patterns of brain atrophy' (i.e., SuStaln trajectory) in each subgroup, separately. We found there was a similar pattern of the three trajectories (left hippocampus-led, right hippocampus-led and cortex-led) in the two disease subgroups (Supplementary Figure S7). This result suggests that the distinct spatiotemporal patterns of brain atrophy may not be affected by disease progress.

Figure S7. Spatiotemporal trajectories of brain atrophy (visualized by positional variance diagrams) in a short-term subsample ($n=148$, mean disease duration= 4.8 ± 2.7 years) and a long-term subsample ($n=148$, mean disease duration= 17.5 ± 7.4 years), separately. Positional variance diagrams visualize the cumulative probability that each region of interested (ROI) has reached a particular z-score (1, 2 or 3) labelled by three colors. The color indicates the level of severity of gray matter reduction: red is mildly affected (z-score=1, i.e., 1 standard deviation unit from healthy control average); purple is moderately affected (z-score=2); and blue is severely affected (z-score=3).

Ref 1/3

In line 373, the authors mentioned that they multiplied z-scores by -1 so that the z-score increases as regional thickness/volume decreases. This is a bit confusing when looking at Figures, for example, Figure S3: “ROI-wise z score of bilateral amygdala volume. Note that a negative z-score represents a larger gray matter volume relative to the healthy control group.” Is it a larger volume or more atrophy? Please clarify.

Response: We thank for the comment. According to the reviewer’s suggestion, we revised the Figure S3 (which is Extended Figure 1 in the revised manuscript) as follows. As a negative z-score represents an increased gray matter volume relative to the healthy control group, we flipped the Y-axis in the bar graph so that a taller bar corresponds to a larger volume for visualization.

Extended Figure 1. ROI-wise z score of bilateral amygdala volume. Note that a negative z-score represents a larger gray matter volume relative to the healthy control group. The dashed line indicates the average of the healthy control population (i.e. $z=0$). The volumes of bilateral amygdala are larger in subtype 4 compared to other subtypes. Error bar indicates standard error.

Reviewer #2 (Remarks to the Author)

Jiang, Li, and coauthors employ the Subtype and Stage Inference (SuStaln) algorithm to discover subtypes of temporal lobe epilepsy (TLE) based on patterns of brain atrophy (or lack thereof). The authors find three different atrophy patterns among TLE patients, plus a subgroup with no atrophy. Interestingly, the no-atrophy subgroup showed greater amygdala volume compared to controls and showed better 5-year treatment response to medication (as opposed to surgery), while the three atrophy subtypes showed better response to surgery.

The findings in this study are exciting examples of how data-driven approaches can identify measurable biological features that help distinguish subgroups of individuals that vary along clinically-relevant dimensions. The 5-year follow-up data in this study, validation sample, and relatively large sample size are great strengths of this study. However, there are flaws in the use and interpretation of SuStaln, limiting the interpretability of certain findings, while other potentially interesting findings are underdeveloped. Details follow. Note that I am an expert in the SuStaln algorithm, but am not an expert in TLE.

Response: We thank the reviewer for their positive appraisal of our work and we are grateful for the detailed feedback focused on a few key concerns. As anticipated by the reviewer, addressing these issues rigorously and comprehensively has entailed major additional analyses, which have now been included in the paper as described in more detail below.

[BLACK] - ORIGINAL COMMENT

[BLUE] - RESPONSE TO COMMENT

[HIGHLIGHTED] - NEW TEXT AND FIGURE/TABLE CHANGES

METHODOLOGICAL CONCERNS

1) The SuStaln input information is not adequately described. The authors should disclose what z-score thresholds are used as “waypoints” in the SuStaln model (Young 2018 Nat Comms). It is also not possible to infer this from the figures. The z-score range 0-2 in Figure 1 but 0-3 in Figure 2. Why this discrepancy?

Response: We thank the reviewer for pointing this issue. We apologize for not describing this information. In fact, we used the z-score thresholds ($z=1, 2, 3$) as “waypoints” in the SuStaln model. We added this important information to the revised manuscript. We also thanks the reviewer for pointing the discrepancy of color bar displaying in Figure 1 and Figure 2. We have revised them to keep consistency.

<<The following changes have been made to the Main Text>>

In *Methods 4.4 Subtype and Stage Inference (SuStaln)*

We used the z-score thresholds ($z=1, 2, 3$) as “waypoints” in the SuStaln model [18].

2) The study is a bit underpowered as the feature to sample ratio is quite low. Assuming $z=1,2,3$ (but I'm guessing here, see above), that's 3 temporal features and 25 regional features, totaling 75 features. With $n=296$ subjects, that's slightly less than 4 subjects per feature, which is not ideal. The authors should ask themselves if all 25 regional features are necessary. Do they all differ between controls and patients (if not their inclusion is not necessary)? Is it possible to combine features or reduce dimensionality in some way? If not, the authors should at least acknowledge being somewhat underpowered in the limitations.

Response: We thank the reviewer for raising the issue. According to the suggestion, we combined the 23 regional features into 13 features by merging regions within the same cortical lobe (**Supplementary Table 7**). Using the 13 features, we re-trained a SuStaln model; the optimal cluster number was still $K=3$. We evaluated the stability of individual subtype label between the original model and new model. We observed that 93.9% of individuals were consistent (**Supplementary Table 5**), indicating a high stability for individual subtype even at relatively fewer spatial features for SuStaln model.

We also acknowledge that there was a bit underpowered as the ratio of sample to feature is low. But we verified the consistency of subtype at a relatively lower but acceptable spatial resolution. This was noted as one of limitations in the revised manuscript.

<<The following changes have been made to the Main Text>>

In *Discussion*

There was a bit underpowered as the ratio of sample to feature is low. But we verified the consistency of subtype at a relatively lower but acceptable spatial resolution.

In *Results 2.3 Reproducibility of SuStaln subtypes*

In addition, we evaluated the stability of SuStaln using different number of features. We observed that 93.9% of individuals were consistent for subtype label (**Supplementary Table 5**), indicating a high stability for individual subtyping even at relatively fewer spatial features for SuStaln model.

In *Results 5.0 Exploratory analysis*

To evaluate the stability of SuStaln at a relative lower spatial resolution, the 23 ROI features were down sampled to 13 features by merging regions within the same cortical lobe (**Supplementary Table 7**).

<<The following changes have been made to the Supplementary Materials>>

Supplementary Table 7. Regions of interest (ROIs) included for SuStain modeling.

The following 23 ROI features are used for initial SuStain modeling		
Left Hippocampus	Right Hippocampus	Right Pallidum
Left Putamen	Left Thalamus	Right Thalamus
Left Caudal Middle frontal gyrus	Right Caudal Middle frontal gyrus	Left Entorhinal gyrus
Left Fusiform gyrus	Left Paracentral gyrus	Right Paracentral gyrus
Left Parahippocampal gyrus	Right Pars opercularis	Right Pars triangularis
Left Precentral gyrus	Right Precentral gyrus	Left Precuneus
Right Precuneus	Left Superior frontal gyrus	Right Superior frontal gyrus
Left Temporal pole	Left Transverse temporal gyrus	
The following 13 ROI features (merged from above 23 ROIs) are used for validation		
Left Hippocampus	Right Hippocampus	Right Pallidum
Left Putamen	Left Thalamus	Right Thalamus
Right Frontal lobe	Right Sensorimotor cortex	Right Parietal lobe
Left Frontal lobe	Left Sensorimotor cortex	Left Parietal lobe
Left Temporal lobe		

Supplementary Table 5. Consistency of individual subtype label for SuStain model based on 23 ROI features and 13 ROI features.

		Individual Subtype (Model_ROI13)			
		Subtype 1	Subtype 2	Subtype 3	Subtype 4
Individual Subtype (Model_ROI23)	Subtype 1	85	0	0	0
	Subtype 2	1	110	2	0
	Subtype 3	3	4	31	3
	Subtype 4	1	3	1	52

3) One aspect of the study that breeds optimism are: point 2 above is the validation analysis in the separate sample. However, reporting of this information is quite sparse. There is no mention of any validation procedures in the Methods (outside of describing the validation sample), and there is only one supplemental figure with hardly any mention. A quantitative comparison between the discovery and validation sets is warranted. Could the authors correlate the (subtype vs control) effect maps? And/or correlate the event position of features between the two datasets?

Response: We thank the reviewer for her/his positive comment regarding to the validation analysis. We also thank the reviewer for suggesting us to conduct a quantitative comparison, and giving us an opportunity to clarify the details of validation procedures. In the revised manuscript, we added two sections (**Methods 4.6** and **Results 2.3**) to clarify the details of validation procedures and results. In addition, we supplemented a quantitatively correlation analysis in **Methods 4.6** (also described as below) between the subtype vs. control effect map (i.e., z-score map) of discovery set and that of validation set. The correlation result shows a high consistency between the paired-wise maps for each subtype (subtype 1: $r=0.855$, $p<10e-10$; subtype 2: $r=0.789$, $p<10e-10$; subtype 3: $r=0.809$,

$p > 10e-10$; subtype 4: $r = 0.705$, $p < 10e-10$) (**Extended Figure 3**).

We revise the manuscript and clarify the details of validation procedures in the revised manuscript as follows.

<<The following changes have been made to the Main Text>>

In Methods 4.6 Reproducibility of SuStaln subtypes in another independent sample

We further examined the reproducibility of the SuStaln trajectories in another independent validation sample including 109 patients diagnosed with temporal lobe epilepsy. Following the same image processing described in Methods 4.3, we extracted ROI-wise z-score for each patient. Subsequently, the SuStaln trajectory was re-estimated based on the validation data using the same SuStaln parameters with the modeling of discovery database (described in **Methods 4.4**). The spatiotemporal trajectory can be mathematically characterized as a sequence of ranked biomarkers (here $n=23$), which is shown in the (Supplementary Table 4). In addition, SuStaln assigned each patient into a subtype, which allowed us to calculate average of z score map across individuals within the same subtype as a representation of subtype-specific atrophy signature (**Figure 2**). Pearson correlation coefficient was used as a quantitative coefficient to evaluate the consistency of z score map between discovery dataset and validation dataset. Spatial autocorrelation in brain map was corrected by a spatial autocorrelation-preserving permutation test (termed 'spin test') [44].

In Results 2.3 Reproducibility of SuStaln subtypes

We examined the reproducibility of the SuStaln trajectories in another independent validation sample including 109 patients diagnosed with temporal lobe epilepsy (61 females, age= 33.1 ± 10.4 years). The SuStaln trajectory was re-estimated based on the validation data. The spatiotemporal trajectory can be mathematically characterized as a sequence of ranked biomarkers (here $n=23$), which is shown in the (Supplementary Table 4). We observed again that the three trajectories from the validation data were began at the left hippocampus, right hippocampus and cortex separately, consistent with the findings from the discovery cohort. In addition, SuStaln assigned each patient into a subtype, which allowed us to calculate average of z score map across individuals within the same subtype as a representation of subtype-specific atrophy signature. Four distinct signatures of brain atrophy patterning were replicated in the validation dataset (**Extended Figure 3**). We observed a high consistency of z score map between discovery dataset and validation dataset ($r > 0.7$, $p < 10e-10$). These results suggested the reproducibility of SuStaln subtypes.

Extended Figure 3. Four distinct neuroanatomical signatures of brain atrophy patterning in people with temporal lobe epilepsy in discovery dataset and validation dataset separately. Pearson correlation coefficient is used to evaluate the consistency of z score map between discovery dataset and validation dataset.

- 4) Some aspects of the subtypes could use to be described in more detail:
 - a) Please include a (preferably cross-validated positional variance diagram so the reader can get a sense of the confidence in each subtype sequences.
 - b) It would be helpful to describe comparisons of subtypes between one another in addition to between subtypes and controls.
 - c) What stages are being visualized in Figure 1A-C?

d) It would help to include a supplemental figure plotting subtype probability across stages, stratified by subtypes. This helps visualize whether there are “cross-over events” — moments in the subtype sequence when two subtypes “merge” (see e.g. Aksman et al., 2023 Brain; Young, Vogel et al., 2023 Brain). These are important to identify because, if present, they make interpretation somewhat more challenging.

Response: We thank the reviewer for these comments. According to the suggestions, we made revisions as follows.

a) We added a positional variance diagram (**Supplementary Figure S3**, also provided as below) to visualize the cumulative probability of each subtype sequence.

Figure S3. Positional variance diagrams visualize the cumulative probability that each region of interested (ROI) has reached a particular z-score (1, 2 or 3) labelled by three colors. The color indicates the level of severity of gray matter reduction: red is mildly affected (z-score=1, i.e., 1 standard deviation unit from healthy control average); purple is moderately affected (z-score=2); and blue is severely affected (z-score=3).

b) We added comparisons of subtypes between one another in addition to between subtypes and controls. We showed these results in **Extended Figure 2** (also provided as below) and **Supplementary Table 3**.

Extended Figure 2. Comparisons of ROI-wise z score between any two subtypes in addition to between subtypes and healthy control group (HC). Color bar indicates T value of two sample t-test with FDR correction. A positive T value (blue) represents a reduction in terms of gray matter cortical thickness or subcortical volume.

c) We added the label of stages which were visualized in **Figure 1A-C** (also provided below).

Figure 1. Spatiotemporal patterns of progression of brain atrophy via SuStain. Trajectory shows that cortical thickness or volume loss is firstly observed in the left hippocampus (a), the right hippocampus (b) and cortex (c) in people with temporal lobe epilepsy relative to healthy controls. The color of brain region reveals the severity of grey matter loss; white: unaffected areas ($z < 1$); light blue: mildly affected areas ($z = 1-2$); dark blue: severely affected areas ($z > 2$). (d) Individual subtyping according to the maximum probability of belonging to which 'trajectory' (red, left hippocampus-predominant trajectory; blue, right-hippocampus-predominant trajectory; green, cortex-predominant trajectory). Note that individuals who do

not deviant obvious reduction in any regions were assigned into a single subgroup (orange, 'normal' signature). **(e-g)** Correlation between SuStaln stages and z scores (i.e., the degree of thickness/volume decrease in patients relative to healthy population) of average cortical thickness, the volume of left and right hippocampus separately in each subgroup (red, left hippocampus-predominant trajectory; blue, right-hippocampus-predominant trajectory; green, cortex-predominant trajectory). **p<0.001, *p<0.05.

d) We added a supplemental figure (**Supplementary Figure S4**, also provided below) plotting subtype probability across stages, stratified by subtypes. This figure shows that the probability of maximum likelihood subtype is high across all SuStaln stage, indicating that there is no "cross-over events" in the subtype sequence. This is also supported by the result of consistency of SuStaln subtype label in a longitudinal subsample having followed up MRI data. To examine whether the subtype label keeps consistency as disease progresses, we followed up brain MRI data of a subsample (n=23, average of interval time=39.0±16.8 months). The labels of subtype at follow-up remained consistent with baseline for almost all patients (**Supplementary Figure S6**), suggesting that since certain initial brain injury is established, it is less likely to shift from one trajectory pattern (i.e., subtype) to another.

Figure S4. Probability of maximum likelihood trajectory is high across almost all SuStaln stages.

Figure S6. Subtype labels remain consistent for almost all patients at baseline and follow-up.

<<The following changes have been made to the Main Text>>

In *Methods 4.4 Subtype and Stage Inference (SuStaln)*

The cumulative probability for each ROI to reach a particular z-score over SuStaln stage is visualized using a positional variance diagram (Supplementary Figure S3).

In *Results 2.2 Subtype-specific neuroanatomical signatures*

In addition, comparisons of ROI-wise z score between any two subtypes are visualized in Extended Figure 2. Results of inter-subtype comparison that includes all ROIs across the brain are described in Supplementary Table 3.

Extended Figure 2. Comparisons of ROI-wise z score between any two subtypes in addition to between subtypes and healthy control group (HC). Color bar indicates T value of two sample t-test with FDR correction. A positive T value (blue) represents a reduction in terms of gray matter cortical thickness or subcortical volume.

<<The following changes have been made to the Supplementary Materials>>

ROI label
ROI 01: Thalamus.L
ROI 02: Putamen.L
ROI 03: Hippocampus.L
ROI 04: Thalamus.R
ROI 05: Pallidum.R
ROI 06: Hippocampus.R
ROI 07: Caudalmiddlefrontal.R
ROI 08: paracentral.R
ROI 09: Parsopercularis.R
ROI 10: Parstriangularis.R
ROI 11: Precentral.R
ROI 12: Precuneus.R
ROI 13: Superiorfrontal.R
ROI 14: Caudalmiddlefrontal.L
ROI 15: Entorhinal.L
ROI 16: Fusiform.L
ROI 17: Parahippocampal.L
ROI 18: Paracentral.L
ROI 19: Precentral.L
ROI 20: Precuneus.L
ROI 21: Superiorfrontal.L
ROI 22: Temporalpole.L
ROI 23: Transversetemporal.L

Figure S3. Positional variance diagrams visualize the cumulative probability that each region of interested (ROI) has reached a particular z-score (1, 2 or 3) labelled by three colors. The color indicates the level of severity of gray matter reduction: red is mildly affected (z-score=1, i.e., 1 standard deviation unit from healthy control average); purple is moderately affected (z-score=2); and blue is severely affected (z-score=3).

Figure S4. Probability of maximum likelihood trajectory is high across almost all SuStaln stages.

5) The authors seem to misunderstand the meaning of Stage 0 individuals. If an individual is stage 0, they do not have enough abnormal feature values to be classified into a subtype. However, SuStaln will assign them subtypes and probabilities anyway. These probabilities are likely meaningless as they are not based on meaningful variation in feature values. Therefore, any analyses involving SuStaln stage or subtype probabilities should exclude Stage 0 (no-atrophy) individuals. This includes e.g. Fig 1 e-g., the logistic regression analysis, and the main findings of Table 1 — how can one covary for stage here when one of the “subtypes” always has a stage of 0?

Response: Thanks for the comment. We re-performed analyses involving SuStaln stage or subtype probabilities after excluding individuals with a stage 0. We replaced by these new results as follows.

The new correlation results between SuStaln stages and regional z scores were provided in (**Fig 1e-g, Supplementary Table 1**) and also described briefly here. There was a significant correlation between SuStaln stages and average cortical thickness (Figure. 1e, trajectory 1: $r=0.599$, $p<0.001$; trajectory 2: $r=0.791$, $p<0.001$; trajectory 3: $r=0.847$, $p<0.001$), as well as the volume of the left hippocampus (Figure. 1f, trajectory 1: $r=0.627$, $p<0.001$; trajectory 2: $r=0.577$, $p<0.001$; trajectory 3: $r=0.431$, $p=0.005$). The significant correlation between SuStaln stages and right hippocampus volume was only found in the 'trajectory' 3 (Figure. 1g, $r=0.269$, $p=0.013$). These findings suggest that the SuStaln stage may reflect the underlying neurophysiological and pathological processes.

The main results of Table 1 do not need to be modified because they are based on comparisons between subtypes, and do not involve any analysis of SuStaln stage or subtype probabilities.

<<The following changes have been made to the Main Text>>

In Results 2.2 Subtype-specific neuroanatomical signatures

Specifically, there was a significant correlation between SuStaln stages and average cortical thickness (**Figure. 1e**, trajectory 1: $r=0.599$, $p<0.001$; trajectory 2: $r=0.791$, $p<0.001$; trajectory 3: $r=0.847$, $p<0.001$), as well as the volume of the left hippocampus (**Figure. 1f**, trajectory 1: $r=0.627$, $p<0.001$; trajectory 2: $r=0.577$, $p<0.001$; trajectory 3: $r=0.431$, $p=0.005$). The significant correlation between SuStaln stages and right hippocampus volume was only found in the 'trajectory' 3 (**Figure. 1g**, $r=0.269$, $p=0.013$). These findings suggest that the SuStaln stage may reflect the underlying neurophysiological and pathological processes. **Supplementary Table 1** provides ROI-wise correlation coefficients between SuStaln stages and regional z scores.

Figure 1. Spatiotemporal patterns of progression of brain atrophy via SuStaln.

Trajectory shows that cortical thickness or volume loss is firstly observed in the left hippocampus (a), the right hippocampus (b) and cortex (c) in people with temporal lobe epilepsy relative to healthy controls. The color of brain region reveals the severity of grey matter loss; white: unaffected areas ($z < 1$); light blue: mildly affected areas ($z = 1-2$); dark blue: severely affected areas ($z > 2$). (d) Individual subtyping according to the maximum probability of belonging to which ‘trajectory’ (red, left hippocampus-predominant trajectory;

blue, right-hippocampus-predominant trajectory; green, cortex-predominant trajectory). Note that individuals who do not deviate obvious reduction in any regions were assigned into a single subgroup (orange, 'normal' signature). **(e-g) Correlation between SuStain stages and z scores (i.e., the degree of thickness/volume decrease in patients relative to healthy population) of average cortical thickness, the volume of left and right hippocampus separately in each subgroup (red, left hippocampus-predominant trajectory; blue, right-hippocampus-predominant trajectory; green, cortex-predominant trajectory). **p<0.001, *p<0.05.**

<<The following changes have been made to the Supplementary Materials >>

Supplementary Table 1. ROI-wise correlation coefficients between SuStain stages and regional z scores.

Features	Trajectory 1 (n=85)		Trajectory 2 (n=113)		Trajectory 3 (n=41)	
	r	p	r	p	r	p
Mean_of_cortical_regions	.599**	0.000	.791**	0.000	.847**	0.000
L_thalamus	.659**	0.000	.598**	0.000	.404*	0.009
L_putamen	.419**	0.000	.537**	0.000	-0.022	0.890
L_hippocampus	.627**	0.000	.577**	0.000	.431*	0.005
R_thalamus	.452**	0.000	.426**	0.000	0.17	0.287
R_pallidum	.373**	0.000	.513**	0.000	0.256	0.106
R_hippocampus	.269*	0.013	0.157	0.097	-0.006	0.973
R_caudalmiddlefrontal	.446**	0.000	.745**	0.000	.598**	0.000
R_paracentral	.464**	0.000	.656**	0.000	.685**	0.000
R_parsopercularis	.462**	0.000	.621**	0.000	.367*	0.018
R_parstriangularis	.417**	0.000	.611**	0.000	.540**	0.000
R_precentral	.491**	0.000	.686**	0.000	.684**	0.000
R_precuneus	.415**	0.000	.729**	0.000	.810**	0.000
R_superiorfrontal	.456**	0.000	.679**	0.000	.759**	0.000
L_caudalmiddlefrontal	.455**	0.000	.710**	0.000	.721**	0.000
L_entorhinal	.499**	0.000	.458**	0.000	.545**	0.000
L_fusiform	.580**	0.000	.664**	0.000	.687**	0.000
L parahippocampal	.485**	0.000	.462**	0.000	.692**	0.000
L_paracentral	.393**	0.000	.713**	0.000	.515**	0.001
L_precentral	.517**	0.000	.689**	0.000	.753**	0.000
L_precuneus	.590**	0.000	.675**	0.000	.787**	0.000
L_superiorfrontal	.507**	0.000	.683**	0.000	.558**	0.000
L_temporalpole	.458**	0.000	.425**	0.000	.406**	0.009
L_transversetemporal	.236*	0.029	.385**	0.000	.657**	0.000

**p<0.001, *p<0.05

6) There are several issues with the “sp-score” analysis, as follows:

- a) It is important to note this is not a normally distributed variable (not even close), which is a consideration in the use of linear models.
- b) My understanding is that the “sp-score” is just a rebranding of the maximum-likelihood subtype probability. If so, the authors may misunderstand what this variable is. The LR analysis is essentially just predicting how stably a subject belongs to *their mostly likely* subtype. This analysis can be thought of as establishing which non-MR variables could contribute to subtype confidence. But this analysis would also change depending on which subtype probability is being used — when using the maximum-likelihood probability, different information sources are being mixed together. It would make more sense to perform the LR on each subtype probability separately.
- c) Related: On multiple occasions, the author claim the sp-score is “interpreted” as something, based on the LR analysis. That doesn’t make sense. The sp-score already has a very clear interpretation based on the SuStaln model formulation, and it is entirely to do with how much an individual’s MR pattern matches that of their maximum likelihood canonical subtype. So, in this analysis, the authors are merely describing things associated with subtype probability
- d) How were the features for the sp-score analysis chosen? Why isn’t SuStaln stage included?
- e) The logistic regression approach used is likely optimistically biased, and does not qualify as “AI”. Was there any cross-validation? Parameter tuning? Was a linear kernel used? The details are sparse but, without them, the reader assumes this model is overfit.
- f) Finally, this analysis is generally oversold and sensationalized. On line 282 the authors claim to have “introduced” this metric, when it is a main output of the SuStaln algorithm and is commonly used across all SuStaln papers, either as a means of exclusion (e.g. Collij et al., 2022 Neurology) or to compare subtype confidence with other measures (e.g. Vogel et al., 2021 Nat Med, Young et al., 2023 Brain).

Response: We thank the reviewer for pointing this issue. According to the suggestions, we made revisions. We agree with the reviewer’s comment that the “sp-score” is a rebranding of the maximum-likelihood subtype probability. We acknowledge that the “sp-score” already has a very clear interpretation based on the SuStaln model formulation. The motivation doing such sp-score analysis was to make it easier for clinical doctors or other non-AI researchers to easily understand the potential meaning/association of the indicator (classification probability) with other variables. To this purpose, we investigated which non-MR variables could contribute to subtype confidence (i.e., sp-score) in each subtype. Specifically, we used Spearman rank correlation analysis (sp-score is not normally distributed) to examine the association between sp-score and clinical variables. The results are shown in following table.

Table R3. Association between sp-score and clinical or MRI variables.

	Age of Onset	Illness Duration	Left Hippocampus z-score	Right Hippocampus z-score	Right cortex thickness z-score	Left cortex thickness z-score
Subtype 1 sp-score	0.025	-0.015	-0.416**	0.770**	-0.03	-0.161**

Subtype 2 sp-score	-0.171**	0.109	0.522**	-0.638**	-0.432**	-0.305**
Subtype 3 sp-score	0.252**	-0.177**	-0.469**	-0.507**	0.267**	0.253**

** p<0.01

However, we realize that the above result of sp-score analysis (i.e., association between sp-score and clinical variables in each subtype) is similar with the result in 2.4 Clinical characterization of subtypes. Thus, we decide not to report this replicated result in the main text; instead, we replace with a new prediction analysis to identify treatment outcomes (Method 4.9). Specifically, we re-conducted a machine-learning analysis with or without SuStaln subtype (instead of sp-score) included as a prior background, to measure its contribution to treatment-outcome prediction. The main part of prediction analysis is described in following Response to Q10. Accordingly, we have also revised some of the discussion and conclusions.

ANALYTIC / CONCEPTUAL CONCERNS

7) I was surprised and fascinated to see a subtype effect on total intracranial volume — individuals with the cortical subtype apparently had significantly larger crania than the other three subtypes. Given important recent associations between head size and disease (e.g. Seidlitz et al., 2022 bioRxiv; Adams et al., 2016 Nat Neuro), it is of great interest to establish whether this finding is biological or methodological. Was TIV different between that subtype and controls? Can the authors plot this particular association? Also, if there is a subtype difference in TIV, does that complicate the fact that authors must control volumetric effects by TIV? The TIV finding is not mentioned at all in the results or discussion, but I believe it deserves attention.

Response: Thanks for the comment. This is indeed a very interesting point regarding TIV difference among subtypes. We think that this finding is not a methodological effect. Firstly, the TIV and other non-interest factors (sex, age) have been controlled by using a regression model and z score transformation (see Methods 4.3 Image processing). Subsequently, adjusted z scores were used for SuStaln modeling. In addition, as the brain structural difference between subtypes may be affected by TIV, we have regressed out the volumetric effect (TIV) before ROI-wise z score comparisons.

To clarify TIV's influence on clinical features, we also conducted the following analyses.

a) We compared the TIV difference between subtypes and healthy controls (HC) by ANOVA. Post-hoc comparisons (Bonferroni method) were also conducted to compare difference between any of two subgroups. We found that compared with HC, the TIV was significantly smaller in subtype 1 (p<0.001), subtype 2 (p=0.004) and subtype 4 (p<0.001), but larger in subtype 3 (p=0.036). This comparison is shown in following figure S1-1.

Figure S1-1. Comparisons of TIV among subtype 1, 2, 3, 4 and HC.

b) To examine whether TIV was associated with specific clinical features, we investigated the correlation between TIV and age of onset as well as illness duration using Pearson correlation analysis. We found that there was no significant correlation between TIV and age of onset ($r=0.028$, $p=0.632$) or illness duration ($r=-0.107$, $p=0.066$) in the patient group. In addition, we compared the difference of TIV between clinical subgroups using ANOVA. We found that the TIV was significantly smaller in the HS subgroup compared to the non-HS subgroup ($F=7.28$, $p=0.007$). The FS seizure type subgroup showed smaller TIV than the BECT seizure type subgroup ($F=4.03$, $p=0.045$). In addition, the male patient group showed larger TIV than the female patient group ($F=53.5$, $p<0.001$). There was no significant difference of TIV between clinical subgroups ($p>0.05$) in term of other clinical features, including seizure lateralization, history of hypoxia at birth, history of head trauma, history of febrile seizures, history of encephalitis meningitis, history of positive family, aura, seizure frequency, seizure type, medications, pathology waves and treatment outcomes. These results indicated that TIV may not drive the differences of clinical features between-subtypes, although the subtype 3 showed a larger TIV than other subtypes.

In the revised manuscript, we added a part to mention above findings in the exploratory analyses.

<<The following changes have been made to the Main Text>>

In Results 2.4 Clinical characterization of subtypes

We also found a subtype effect on total intracranial volume (TIV) — individuals with the cortical subtype 3 had significantly larger intracranial volume than the other three subtypes; an exploratory analysis was used to examine the association of TIV with subtypes and clinical features.

In Methods 5.0 Exploratory analysis

We investigated the difference of total intracranial volume (TIV) between subtypes and healthy controls; we also examine whether TIV was associated with specific clinical features (Supplementary Materials).

<<The following changes have been made to the Supplementary Materials >>

In Method S1 Total intracranial volume (TIV) analyses

We found a subtype effect on total intracranial volume (TIV) — individuals with the cortical subtype (subtype 3) apparently had significantly larger TIV than the other three subtypes. To clarify it, we conducted the following analyses.

1) We compared the TIV difference between subtypes and healthy control group (HC) by ANOVA; post-hoc comparisons (Bonferroni method) were also conducted to compare difference between any of two subgroups. We found that compared with HC, the TIV was significantly smaller in subtype 1 ($p<0.001$), subtype 2 ($p=0.004$) and subtype 4 ($p<0.001$), but larger in subtype 3 ($p=0.036$). This comparison is shown in following figure S1-1.

Figure S1-1. Comparisons of TIV among subtype 1, 2, 3, 4 and HC.

2) To examine whether TIV was associated with specific clinical features, we investigated the correlation between TIV and age of onset as well as illness duration using Pearson correlation analysis. We found there was no significant correlation between TIV and age of onset ($r=0.028$, $p=0.632$) or illness duration ($r=-0.107$, $p=0.066$) in the patient group. In addition, we compared the difference of TIV between clinical subgroups using ANOVA. We found that the TIV was significantly smaller in the HS subgroup compared to the non-HS subgroup ($F=7.28$, $p=0.007$). The FS seizure type subgroup showed smaller TIV than the BECT seizure type subgroup ($F=4.03$, $p=0.045$). In addition, the male patient group showed larger TIV than the female patient group ($F=53.5$, $p<0.001$). Besides, there was no significant difference of TIV between clinical subgroups ($p>0.05$) in term of other clinical features, including seizure lateralization, history of hypoxia at birth, history of head trauma,

history of febrile seizures, history of encephalitis meningitis, history of positive family, aura, seizure frequency, seizure type, medications, pathology waves and treatment outcomes. These results indicated that TIV may not drive the differences of clinical features between-subtypes, although the subtype 3 showed a larger TIV than other subtypes.

8) Are any of the main variables in Table 1 associated with SuStaln stage? Were there any Subtype x Stage interactions?

Response: Thanks for the comment. The reviewer also raises an important point that whether there are any Subtype x Stage interactions. According to the reviewer's suggestion, we conducted an exploratory analysis to investigate whether there are any subtype x stage interaction in clinical features. We used a logistic regression analysis for Table1 categorical Y variables (or linear regression analysis for continuous Y variables) with Subtype (X1), Stage (X2) and Subtype x Stage (X3) as independent variables. These individuals with stage=0 were not included in above analyses. We found that only in seizure lateralization, there was a significant Subtype x Stage interaction effect (beta=0.052, uncorrected p=0.045 but not significant after multiple comparison correction); there was also a significant subtype effect (beta=-2.195, p<0.001). Besides, we found a significant subtype effect in the HS (beta=-1.707, p<0.001), onset age (F=6.756, p<0.001) and illness duration (F=3.85, p=0.023). However, there was no significant interaction effect of Subtype x Stage on other variables (p>0.05) in Table1. We did not find any stage effect on these clinical variables (p>0.05).

9) The most important finding of the study is the subtype effect on treatment response. However, the effect is not so much a subtype effect as an atrophy vs. no atrophy effect. Those without measurable brain atrophy are more likely to respond to medication and those with brain atrophy more likely respond to operation. To establish whether this is truly a subtype effect or a brain atrophy effect, it would be helpful for the authors to plot likelihood of treatment response *across* stage *within* subtype. This can be done with a sliding window approach. Is it the case that the more atrophy a subject has, the more likely they respond to operation (and/or don't respond to medication)? Or is purely a subtype effect where the results don't vary across stage, only across atrophy vs no atrophy subtypes?

Response: Thanks for the comment. According to the reviewer's suggestion, we provided table_R9.1-8 (provided as below) showing number of patient with effective response (medication or surgery) or not, across stages within subtype 1, 2, 3 and a union set (subtype 1 & subtype 2 & subtype 3).

We also calculated the effective rates using a sliding window-based approach. Specifically, window length was defined as three adjacent stages, and each sliding step was set to one stage. Effective rate was defined as the number of people in the window divided by the total number of people in the window. Effective rate of each window was calculated as the window slides (also provided in following tables R9.9, R9.10). According to the sliding-

window response rate, it seems that the results don't vary across stages. However, this phenomenon still needs to be considered with caution due to the sample size. We must acknowledge that the current sample size is hard to plot a trajectory showing how treatment response rate changes as stages/atrophy increases. This point is also added as one of limitations of this study.

Table R9.1 Number of patients respond to medication treatment in subtype 1 across SuStaln stages

Subtype 1	SuStaln Stage									
	1	2	3	4	5	6	7	8	14	33
Effective	0	0	0	1	0	0	0	0	0	1
Ineffective	3	5	2	4	2	1	1	1	1	0
Total	3	5	2	5	2	1	1	1	1	1

Table R9.2 Number of patients respond to medication treatment in subtype 2 across SuStaln stages

Subtype 2	SuStaln Stage														
	1	2	3	4	5	6	7	10	13	14	18	19	20	22	27
Effective	0	0	3	0	1	0	0	0	0	0	0	1	0	0	0
Ineffective	2	3	10	2	4	1	2	2	1	1	1	0	1	2	1
Total	2	3	13	2	5	1	2	2	1	1	1	1	1	2	1

Table R9.3 Number of patients respond to medication treatment in subtype 3 across SuStaln stages

Subtype 3	SuStaln Stage												
	3	4	5	6	7	9	10	12	14	15	20	27	38
Effective	0	0	0	0	0	0	2	0	0	1	0	0	0
Ineffective	2	1	1	3	2	1	0	1	2	0	1	1	1
Total	2	1	1	3	2	1	2	1	2	1	1	1	1

Table R9.4 Number of patients respond to medication treatment in a union set across SuStaln stages

Subtype 1&2&3	SuStaln Stage																					
	1	2	3	4	5	6	7	8	9	10	12	13	14	15	18	19	20	22	27	33	38	
Effective	0	0	3	1	1	0	0	0	0	2	0	0	0	1	0	1	0	0	0	0	1	0
Ineffective	5	8	14	7	7	5	5	1	1	2	1	1	4	0	1	0	2	2	2	2	0	1
Total	5	8	17	8	8	5	5	1	1	4	1	1	4	1	1	1	2	2	2	2	1	1

Table R9.5 Number of patients respond to surgery treatment in subtype 1 across SuStaln stages

Subtype 1	SuStaln Stage												
	1	2	3	4	5	6	8	9	10	11	16	20	
Effective	5	7	9	6	4	0	0	1	2	1	1	0	
Ineffective	1	4	2	2	1	1	2	0	2	0	0	1	
Total	6	11	11	8	5	1	2	1	4	1	1	1	

Table R9.6 Number of patients respond to surgery treatment in subtype 2 across SuStaln stages

Subtype 2	SuStaln Stage																
	1	2	3	4	5	6	7	10	11	12	14	16	18	19	22	23	31
Effective	3	8	9	2	4	4	1	1	1	2	1	1	0	0	1	0	1

Ineffective	0	3	4	1	2	2	0	0	0	0	0	0	1	1	0	1	0
Total	3	11	13	3	6	6	1	1	1	2	1	1	1	1	1	1	1

Table R9.7 Number of patients respond to surgery treatment in subtype 3 across SuStaln stages

Subtype 3	SuStaln Stage											
	2	3	5	7	9	11	12	20	22	23	28	31
Effective	1	4	0	0	1	2	1	0	1	1	1	0
Ineffective	0	0	1	1	0	0	2	2	0	0	0	1
Total	1	4	1	1	1	2	3	2	1	1	1	1

Table R9.8 Number of patients respond to surgery treatment in a union set across SuStaln stages

Subtype 1&2&3	SuStaln Stage																				
	1	2	3	4	5	6	7	8	9	10	11	12	14	16	18	19	20	22	23	28	31
Effective	8	16	22	8	8	4	1	0	2	3	4	3	1	2	0	0	0	2	1	1	1
Ineffective	1	7	6	3	4	3	1	2	0	2	0	2	0	0	1	1	3	0	1	0	1
Total	9	23	28	11	12	7	2	2	2	5	4	5	1	2	1	1	3	2	2	1	2

Table R9.9 Effective rate of each window respond to medication treatment across sliding windows

Medication	Sliding windows																			
	1	2	3	4	5	6	7	8	9	10	11	12	13	14	15	16	17	18	19	
Subtype 1	0.00	0.08	0.11	0.13	0.00	0.00	0.00	0.33	-	-	-	-	-	-	-	-	-	-	-	-
Subtype 2	0.17	0.17	0.20	0.13	0.13	0.00	0.00	0.00	0.00	0.33	0.33	0.25	0.00	-	-	-	-	-	-	-
Subtype 3	0.00	0.00	0.00	0.00	0.40	0.50	0.40	0.25	0.25	0.33	0.00	-	-	-	-	-	-	-	-	-
Subtype 1&2&3	0.10	0.12	0.15	0.10	0.06	0.00	0.00	0.33	0.33	0.33	0.00	0.17	0.17	0.67	0.25	0.20	0.00	0.20	0.25	

Table R9.10 Effective rate of each window respond to surgery treatment across sliding windows

Surgery	Sliding windows																			
	1	2	3	4	5	6	7	8	9	10	11	12	13	14	15	16	17	18	19	
Subtype 1	0.75	0.73	0.79	0.71	0.50	0.25	0.43	0.67	0.67	0.67	-	-	-	-	-	-	-	-	-	-
Subtype 2	0.74	0.70	0.68	0.67	0.69	0.75	1.00	1.00	1.00	1.00	0.67	0.33	0.33	0.33	0.67	-	-	-	-	-
Subtype 3	0.83	0.67	0.33	0.75	0.67	0.43	0.33	0.50	1.00	0.67	-	-	-	-	-	-	-	-	-	-
Subtype 1&2&3	0.77	0.74	0.75	0.67	0.62	0.45	0.50	0.56	0.82	0.71	0.80	0.75	0.75	0.50	0.00	0.33	0.43	0.80	0.60	

<<The following changes have been made to the Main Text>>

In Discussion

In addition, the current sample size is not enough to characterize a trajectory showing how treatment response changes as atrophy stage increases.

10) Treatment response is an extremely clinically relevant outcome measure. The authors

have shown subtype effects that are very interesting, but it would be helpful to better understand the degree to which this information aids prediction of such outcomes. Rather than the sp-score analysis (which I'm not sure adds much to the study), could the authors perform an LR analysis predicting treatment outcome? I would be interested to see either an inferential analysis where the independent contribution of subtype is quantified compared to other variables, or a machine-learning analysis with and without subtype included as a feature to measure its contribution to treatment-outcome prediction.

Response: We thank the reviewer for their positive appraisal of treatment relevant results. We also realized that it is necessary to better understand the degree to which the subtype information aids prediction of surgery outcomes. To achieve it, we conducted a machine-learning analysis with or without SuStaln subtype included as a prior background, to measure its contribution to treatment-outcome prediction. Specifically, we evaluated prediction performance on classifying the subject who achieves seizure freedom (OG+) or not (OG-) after surgery, using a classical machine learning prediction procedure (see **Methods**). To examine whether the SuStaln subtype information could help to improve prediction, we conducted prediction procedures through a novel framework under SuStaln subtype background (**Extended Figure 4**). Given that each subtype has unique brain structure and clinical characteristics, each subtype may require specific features/classifiers to predict postoperative outcome. Thus, using support vector machine (SVM), we built a specific sub-classifier corresponding to each SuStaln subtype, yielding a classifier cluster to predict treatment outcomes. By leave-one subject-out cross-validation (LOOCV), we observed an acceptable-to-good prediction performance for each sub-classifier to each SuStaln subtype (**Extended Figure 5**); yielding an overall accuracy (71.72%), specificity (80.21%) and sensitivity (55.10%) on the test data. As a comprehensive evaluation, the Youden Index for the SuStaln subtype-based classifier ($J=0.353$) on test data was significantly higher than randomly predictions by permutation test ($p<0.001$) (**Supplementary Figure S1**). As a reference, we also conducted a predictive test without any SuStaln subtype information as prior. Specifically, SVM classifier was trained using clinical information at baseline as features. By LOOCV, we observed 68.27% accuracy, 87.50% specificity and very low sensitivity (30.61%) on the test data; while Youden Index ($J=0.181$) did not show significant difference compared to randomly predictions by permutation test ($p=0.19$).

Taken together, these results suggest that these (OG-) patients were not successfully predicted if only clinical information was relied upon; however, they were successfully predicted if using SuStaln subtype as a prior background (i.e., a classifier cluster with specific sub-classifier to each SuStaln subtype).

<<The following changes have been made to the Main Text>>

In Results 2.5 Subtype-based classifier predicts surgery prognosis

We evaluated prediction performance on classifying the subject who achieves seizure freedom (OG+) or not (OG-) after surgery, using a classical machine learning prediction procedures (see **Methods**). To examine whether the SuStaln subtype information could

help to improve prediction, we conducted machine learning prediction procedures through a novel framework under SuStaln subtype background (Extended Figure 4). We proposed a perspective that each subtype may require specific features/classifiers to predict postoperative outcome, given that each subtype has specific brain structure and clinical characteristics. Thus, using support vector machine (SVM), we built a specific sub-classifier corresponding to each SuStaln subtype. By leave-one subject-out cross-validation (LOOCV), we observed an acceptable-to-good prediction performance for each sub-classifier to each SuStaln subtype (Extended Figure 5); yielding an overall accuracy (71.72%), specificity (80.21%) and sensitivity (55.10%) on the test data. As a comprehensive evaluation, the Youden Index for the SuStaln subtype-based classifier ($J=0.353$) on test data was significantly higher than randomly predictions by permutation test ($p=0.005$) (Supplementary Figure S1). Details of prediction performance of each subtype classifier are described in Supplementary Table S6.

As a reference, we also conducted a predictive test without any SuStaln subtype information as prior. Specifically, SVM classifier was trained using clinical information at baseline as features. By LOOCV, we observed 68.27% accuracy, 87.50% specificity and very low sensitivity (30.61%) on the test data; while Youden Index ($J=0.181$) did not show significant difference compared to randomly predictions by permutation test ($p=0.22$). This suggests that these (OG-) patients were not successfully identified if only clinical information was relied upon. In addition, we found that even if we added much more features (clinical variables + MRI regional measures) to train classifier, the prediction performance did not improve (Youden Index=0.209, accuracy=66.21%, sensitivity=42.86%, specificity=78.13%).

In Methods 4.9 Predicting prognosis of surgery by SuStaln subtype-based prediction model

To examine whether the SuStaln subtype information at baseline could help to predict the prognosis of surgery at follow-up for a given patient, we conducted machine learning procedures to predict treatment outcome in a sub-sample of 145 post-surgery follow-up subjects. Here, we described how to train a support vector machine (SVM) classifier (Supplementary Figure S5). Specifically, we applied leave-one subject-out cross-validation (LOOCV) to obtain train data and test data. In each LOOCV loop, one subject was used as a test set, and the remaining subjects were used as a training set. In training set, the classifier features included the baseline clinical variables, MRI variables, or both. Principal component analysis (PCA) was used to reduce feature dimension. The first N principal components (PCs), which explained beyond 95% of the variance of all features, were used to train a SVM classifier to classify the subject who achieves seizure freedom (OG+) or not (OG-) after surgery. Three commonly used SVM kernel functions (linear, RBF and polynomial) were used. The test set patient's label was predicted based on the built SVM classifier. Prediction performance was measured by sensitivity, specificity and accuracy. We also calculated Youden Index ($\text{sensitivity} + \text{specificity} - 1$) as a comprehensive assessment of both sensitivity and specificity. To further examine whether the prediction

performance is significantly better than random predictions, we used a permutation test to evaluate significance by random permutation of predictive label (Supplementary Materials).

In Discussion

Subtype is a prior important information to aid prediction of surgery outcomes. We built a classifier cluster including specific sub-classifier corresponding to each subtype, which achieved an acceptable-to-good performance on predicting seizure freedom subjects after surgery, better than clinical information-based only prediction model. Although the underlying neural mechanisms are not well understood, we hypothesize that each model requires specific features/classifiers to predict postoperative outcome in subtypes, given that each subtype has unique brain injury and clinical characteristics. This is also supported by previous studies suggesting that TLE patients with certain brain characteristics [29] or clinical features [30] may benefit from temporal surgery. Although there is debate about prognostic factors for surgical outcome in TLE, the presence of hippocampus sclerosis [29, 31], a history of febrile seizure [32] and a low seizure frequency [30] were almost consistently reported to be associated with better outcomes. This novel perspective on building a stratified prediction model may be able to reveal underlying disease heterogeneity in surgery prognosis and guide a more individualized treatment in clinical practice.

Extended Figure 4. (A) A classical prediction model without any SuStaln subtype information. (B) SuStaln subtype-based prediction model with specific sub-classifier corresponding to each SuStaln subtype.

Extended Figure 5. Prediction performance of Machine learning classifier on identifying the subject who achieves seizure freedom or not after surgery. Subtype PC1-4: specific predictive model sub-classifier corresponding to SuStaln subtype 1-4; Subtype PC: an overall classifier cluster of Subtype PC1-4; Reference PC: predictive classifier without any SuStaln subtype information.

<<The following changes have been made to the Supplementary Materials >>

Figure S5. A flowchart of machine learning procedures to predict treatment outcome in a sub-sample of 145 post-surgery follow-up subjects. Leave-one subject-out cross-validation (LOOCV) is used to obtain train data and test data. In each LOOCV loop, one subject is used as a test set, and the remaining subjects are used as a training set. In training set, the classifier features include the baseline clinical variables (C), MRI variables (M), or both (C&M). Principal component analysis (PCA) is used to reduce feature dimension. The first N principal components (PCs), which explained beyond 95% of the variance of all features, are used to train a SVM classifier to classify the subject who achieves seizure freedom (OG+) or not (OG-) after surgery. The test set patient's label is predicted based on the trained SVM classifier. C:

Clinical feature set including age, sex, age of onset, illness duration, seizure lateralization, MRI HS or not, handedness, history of hypoxia at birth, history of head trauma, history of febrile seizures, history of encephalitis meningitis, history of positive family, aura, seizure frequency, seizure type (FS/FBTCs), medications and pathology wave. M: MRI derived regional measures (ROI list in Supplementary Table S7).

MINOR ISSUES

11) The authors first regress out covariates and then z-score data. Typically, one would use a “w-score” technique (La Joie et al. 2012 J Neurosci), where covariates are modeled in controls and then applied to all subjects during the z-scoring procedure. This is the approach used in Young et al. 2018 and most subsequent SuStaln papers.

Response: We thank the reviewer for raising it. We realize that both methods (“z-score” and “w-score”) were used to quantify the extent to which individuals deviate from a population distribution. Both methods have also been applied in the SuStaln modeling studies. Compared to z-scoring procedure, w-score technique requires first building a normative model based on covariates such as age on normal population, and then estimating a value for each patient using the built normative model. Considering our current sample size (n=81) of healthy control groups may not be enough to build a reliable normative model, we did not use w-score method in this study.

Nevertheless, here we also compute the w-score and compare the results obtained using the two methods to see if they are consistent. First, we calculated w-scores of 23 ROI (which would be used as SuStaln features) using the “w-score” technique (La Joie et al. 2012 J Neurosci). Specifically, we used a multiple regression for each ROI to estimate gray matter feature (volume or thickness) as a function of age, square of age, sex and total intracranial volume. Then, for a patient, we entered her/his covariates into that regression model (i.e., normative model) to estimate her/his gray matter feature for a given ROI. The W-score was then calculated as the difference between the estimated gray matter volume/thickness and actual gray matter volume/thickness, divided by the standard deviation of residuals of the normative model fit in the normal population. To compare the consistency between w-score obtained by above methods and z-score adopted in this paper, we calculate the Pearson correlation coefficient (r value) between w-scores and z-scores across the 23 ROIs for each patient. The distribution of r values is displayed in following picture, which shows a high consistency between w-scores and z-scores (the median is $r=0.9433$ across 297 patients).

Figure R11. (a) Frequency diagram showing distribution of r values. (b) Scatter diagram showing Pearson correlation coefficient ($r=0.9433$) between z-scores and w-scores across 23 ROIs of a patient example (id_252).

12) Line 427 — “The statistical comparisons were conducted using ANOVAs with appropriate post-hoc tests for continuous variables...” — Could the authors be more specific?

Response: We thank the reviewer for raising it. We has revised it to ‘The statistical comparisons were conducted using ANOVA with LSD post-hoc tests for continuous variables (age, age of onset, illness duration and TIV) or using Pearson’s Chi-square test for categorical variables. Multiple comparisons were considered with FDR correction.’

<<The following changes have been made to the Main Text>>

In **Methods 4.7 Comparisons of clinical profiles between subtypes**

The statistical comparisons were conducted using ANOVA with post-hoc Least Significant Difference (LSD) tests for continuous variables (age, age of onset, illness duration and TIV) or using Pearson’s Chi-square test for categorical variables. Multiple comparisons were considered with FDR correction.

13) TIV is not necessary as a covariate for cortical thickness (Schwarz et al. 2016 Neuroimage: Clin)

Response: Thanks for the reviewer’s comment. Although previous study has reported that TIV is not necessary as a covariate for cortical thickness (Schwarz et al. 2016, NeuroImage: Clinical), many studies still regard TIV as a non-interest factor and regress it out in epilepsy studies (Whelan et al. 2018, Brain). Thus, TIV was also regress out here to keep data processing consistency with previous studies (Whelan et al. 2018, Brain).

Reference

Schwarz C G, Gunter J L, Wiste H J, et al. A large-scale comparison of cortical thickness and volume methods for measuring Alzheimer's disease severity [J]. *NeuroImage: Clinical*, 2016, 11: 802-812.

Whelan C D, Altmann A, Botia J A, et al. Structural brain abnormalities in the common epilepsies assessed in a worldwide ENIGMA study [J]. *Brain*, 2018, 141(2): 391-408.

14) The authors make a big deal out of the finding of Stage 0 individuals having larger amygdalae compared to controls. But stats on the amygdala (nor hippocampus) are not shown in Figure 3.

Response: Thanks for the comment. Consider that Figure 3 mainly presents the differences in clinical features between subtypes. Therefore, we do not include the amygdala difference in Figure 3. But in the revised manuscript, we highlight the amygdala difference in **Extended Figure 1**.

Extended Figure 1. ROI-wise z score of bilateral amygdala volume. Note that a negative z-score represents a larger gray matter volume relative to the healthy control group. The dashed line indicates the average of the healthy control population (i.e. $z=0$). Error bar indicates standard error.

REVIEWER COMMENTS

Reviewer #1 (Remarks to the Author):

The authors responded to my questions adequately, and the revised manuscript is much improved.

Reviewer #2 (Remarks to the Author):

Jiang, Li and co-authors have made a commendable effort in improving this interesting manuscript. However, there remain issues raised from the previous round of comments, as well as some new issues that have arisen based on new analyses and by providing better transparency. Some of these issues (in particular, #3 below) still strongly affect some of the main findings and interpretations of the study, and therefore must be resolved. See below:

1) In the response to R2 (comment 1, p 13), the authors write “We also thanks the reviewer for pointing the discrepancy of color bar displaying Figure 1 and Figure 2. We have revised them to keep consistency.” However, the issue is still present in the revised manuscript — Figure 1 contains colorbars that range 0-2, but colorbars in Figure 2 (correctly) range from 0-3.

2) I applaud the authors for now including the positional variance diagrams (PVD), which gives much clearer insight into their model and greatly improves transparency (Fig S3). However, this figure is also very revealing in relation to my previous comment #2 concerning power and the number of ROIs. It is quite clear from the PVDs that there is basically no confidence in the ordering of most ROIs — the model is dominated by atrophy of just a few key structures. There is really no real reason to include most of the current ROIs in the model, and doing so needlessly increases model complexity. If the authors reran the model using only relevant regions (e.g. 1,3,4,6,7,17, and just one of any other ROI, keeping in mind its a bit hard to see the numbers), they would probably get the same subtypes but have much more certainty and a less complex model. This is the type of model diagnostics that is needed when using SuStaIn if the authors would ever want this model to generalize outside of the dataset. I recommend the authors re-run the model with these (or a similar list of)

ROIs, which will create a far more interpretable, generalizable and cleaner result.

3) The authors have added a new machine learning analysis predicting treatment response, which represents a definite improvement over the previous analysis. However, this new analysis also contains several important flaws that make it optimistically biased, as follows:

a) Leave-one-out cross-validation is not recommended in machine learning analysis because it is optimistically biased. Instead, please use cross-validation (Varoquaux et al., 2017 Neuroimage; Poldrack et al., 2020 JAMA Neurol).

b) The procedure of first subtyping and then making separate models for each subtype is interesting, but it is once again optimistically biased due to leakage. Specifically, the uncertainty in subtype assignment is not accounted for. To do this properly, the authors would need to fit the a new SuStain model to the training data, and *apply* that model to the left-out data, for each fold of cross-validation. This is again to simulate what would happen in a real setting if this model were to be generalized to new data (the subjects would not already have a subtype).

c) The authors don't have to do this, but to be clear, my original request was a much simpler proposition — to just add subtype (e.g. as a set of dummy variables) and stage to a prediction model, and see if this enhances prediction of treatment outcomes above and beyond a model without the subtype features.

4) I appreciate the new analysis in extended Figure 3 showing correlations between subtype patterns. However, only same subtypes are shown. What is most useful is to show a cross-correlation across all subtypes of the discovery in validation set (as in Vogel et al., 2021 Nat Med; Figure 2B). This is necessary since a) it shows specificity and b) there is usually a baseline level of correlation between atrophy maps due to overall similarity in brain anatomy across people.

5) R1 (comment 1) raises an excellent point stating that it isn't clear whether the atrophy patterns observed are truly "progressive" as would be suggested by the advancement of

stages. The new positional variance diagrams (Fig S3) indeed show that very few subjects advance past stage ~12. In response to this point, the authors provide an interesting and important longitudinal analysis looking at whether subtypes change over time (Fig S6). However, an analysis that would actually answer the Reviewers question more directly would be to chart change in SuStaln *stage* over time. If several subjects show advancing stage over time, this suggests the atrophy is progressive. If not, it may support R1's conjecture that perhaps the atrophy is sporadic.

6) A new study was recently published using a very similar approach and research question to the present study (Xiao et al., 2023 Brain; <https://academic.oup.com/brain/article/146/11/4702/7300937>). The Discussion of this manuscript would definitely be improved by putting the present paper into context with this new study, to describe where findings are convergent, discrepant and complementary.

Manuscript Title: Identification of four biotypes in temporal lobe epilepsy via brain imaging-based machine learning

Yuchao Jiang, Wei Li, et al: Response to Reviewers

Reviewer #1 (Remarks to the Author):

The authors responded to my questions adequately, and the revised manuscript is much improved.

Reviewer #2 (Remarks to the Author):

Jiang, Li and co-authors have made a commendable effort in improving this interesting manuscript. However, there remain issues raised from the previous round of comments, as well as some new issues that have arisen based on new analyses and by providing better transparency. Some of these issues (in particular, #3 below) still strongly affect some of the main findings and interpretations of the study, and therefore must be resolved. See below:

Response: We thank the reviewer for the positive appraisal of our work. We are grateful for the detailed feedback focused on a few key concerns.

[BLACK] - ORIGINAL COMMENT

[BLUE] - RESPONSE TO COMMENT

[HIGHLIGHTED] - NEW TEXT AND FIGURE/TABLE CHANGES

1) In the response to R2 (comment 1, p 13), the authors write “We also thanks the reviewer for pointing the discrepancy of color bar displaying Figure 1 and Figure 2. We have revised them to keep consistency.” However, the issue is still present in the revised manuscript — Figure 1 contains colorbars that range 0-2, but colorbars in Figure 2 (correctly) range from 0-3.

Response: We thanks the reviewer for raising this point. We have replaced the update figures as follows.

<<The following changes have been made to the Main Text>>

Figure 1. Spatiotemporal patterns of progression of brain atrophy via SuStaln. Trajectory shows that cortical thickness or volume loss is firstly observed in the left hippocampus (a), the right hippocampus (b) and cortex (c) in people with temporal lobe epilepsy relative to healthy controls. The color of brain region reveals the severity of grey matter loss; white: unaffected areas ($z < 1$); light blue: mildly affected areas ($z = 1-2$); dark blue: severely affected areas ($z > 2$). (d) Individual subtyping according to the maximum probability of belonging to which 'trajectory' (red, left hippocampus-predominant trajectory; blue, right-hippocampus-predominant trajectory; green, cortex-predominant trajectory). (e-g) Correlation between SuStaln stages and z scores (i.e., the degree of thickness/volume decrease in patients relative to healthy population) of average cortical thickness, the volume of left and right hippocampus separately in each subgroup (red, left hippocampus-predominant trajectory; blue, right-hippocampus-predominant trajectory; green, cortex-predominant trajectory). ** $p < 0.001$, * $p < 0.05$.

Figure 2. Four distinct neuroanatomical signatures of brain atrophy patterning in people with temporal lobe epilepsy. Subtype-specific signature in neuroanatomical pathology includes (1) the left hippocampus-predominant signature (subtype 1), (2) the right hippocampus-predominant signature (subtype 2), (3) the cortex-predominant signature (subtype 3) and (4) the 'normal' signature (subtype 4). ROI-wise z-scores are mapped to a brain template using visualization tools implemented in ENIGMA Toolbox (<https://enigma-toolbox.readthedocs.io/en/latest/index.html>). Color bar indicates z-scores (i.e., normative deviations) relative to the healthy control group. Note that a higher z-score represents a larger gray matter loss. Asterisk indicates significant regional difference between subtype group and healthy control group using two sample t-test following FDR multiple comparisons correction.

2) I applaud the authors for now including the positional variance diagrams (PVD), which gives much clearer insight into their model and greatly improves transparency (Fig S3). However, this figure is also very revealing in relation to my previous comment #2 concerning power and the number of ROIs. It is quite clear from the PVDs that there is basically no confidence in the ordering of most ROIs — the model is dominated by atrophy of just a few key structures. There is really no real reason to include most of the current ROIs in the model, and doing so needlessly increases model complexity. If the authors reran the model using only relevant regions (e.g. 1,3,4,6,7,17, and just one of any other ROI, keeping in mind its a bit hard to see the numbers), they would probably get the same subtypes but have much more certainty and a less complex model. This is the type of model diagnostics that is needed when using SuStaln if the authors would ever want this model to generalize outside of the dataset. I recommend the authors re-run the model with these (or a similar list of) ROIs, which will create a far more interpretable, generalizable and cleaner result.

Response: Thanks for the comments. According to the reviewer's suggestion, we re-run the model with only six ROIs (left thalamus, left hippocampus, right thalamus, right hippocampus, caudalmiddlefrontal cortex, other cortex average). In the new trajectories, the first two subtypes show high confidence in the ordering of six features (**Figure R2.1**). It clearly indicates that the ordering of regional volume loss is left hippocampus to left thalamus to right thalamus to right hippocampus and final to cortex in trajectory 1. In trajectory 2, the ordering is right hippocampus to right thalamus to left thalamus to left hippocampus and final to cortex. In trajectory 3, it shows a non-hippocampus phenotype without clear origin or order. In general, the distinct patterns of three new trajectories are consistent with the original trajectories. Subtype assignment is consistent with original assignment except that fewer patients are assigned to subtype 3 (subtype 1, n=102; subtype 2, n=117; subtype 3, n=19; subtype 4, n=58), compared to original assignments (subtype 1, n=85; subtype 2, n=113; subtype 3, n=41; subtype 4, n=57). This may be due to the inclusion of only two cortical features for the new model. This result suggests that the model needs more cortical features to detect those non-hippocampus-led patients.

Figure R2.1 Positional variance diagrams of six ROI-based SuStaln.

3) The authors have added a new machine learning analysis predicting treatment response, which represents a definite improvement over the previous analysis. However, this new analysis also contains several important flaws that make it optimistically biased, as follows: a) Leave-one-out cross-validation is not recommended in machine learning analysis because it is optimistically biased. Instead, please use cross-validation (Varoquaux et al., 2017 Neuroimage; Poldrack et al., 2020 JAMA Neurol).

Response: Thanks for the professional comment. According to the reviewer’s suggestion, we re-analyze the data using ten-fold cross-validation produce. By ten-fold cross-validation, we observed an acceptable-to-good prediction performance (Extended Figure 5); yielding an overall accuracy (71.72%), specificity (81.03%) and sensitivity (47.87%) on the test data. Youden Index for the SuStaln subtype-based classifier ($J=0.289$) on test data was significantly higher than randomly prediction model by permutation test ($p=0.012$). In contrast, the prediction model without any SuStaln subtype information only shows a poor prediction performance on 89.58% specificity, very low sensitivity (24.49%); while Younden Index ($J=0.141$) did not show significant difference compared to randomly predictions by permutation test ($p=0.307$).

We have replaced the new results as follows.

<<The following changes have been made to the Main Text>>

In **Results 2.5 Subtype-based classifier predicts surgery prognosis**

By ten-fold cross-validation, we observed an acceptable-to-good prediction performance for each sub-classifier to each SuStaln subtype (Extended Figure 5); yielding an overall accuracy (71.72%), specificity (81.03%) and sensitivity (47.87%) on the test data. As a comprehensive evaluation, the Youden Index for the SuStaln subtype-based classifier ($J=0.289$) on test data was significantly higher than randomly predictions by permutation test ($p=0.012$) (Supplementary Figure S1). Details of prediction performance of each subtype classifier are described in Supplementary Table S7

As a reference, we also conducted a predictive test without any SuStaln subtype information as prior. Specifically, SVM classifier was trained using clinical information at baseline as features. By ten-fold cross-validation, we observed 67.59% accuracy, 89.58% specificity and very low sensitivity (24.49%) on the test data; while Youden Index ($J=0.141$) did not show significant difference compared to randomly predictions by permutation test ($p=0.307$). This suggests that these (OG-) patients were not successfully identified if only clinical information was relied upon. In addition, we found that even if we added much more features (clinical variables + MRI regional measures) to train classifier, the prediction performance did not improve (Youden Index=0.130, accuracy=66.90%, sensitivity=24.49%, specificity=88.54%).

In Methods 4.9 Predicting prognosis of surgery by SuStaln subtype-based prediction model

Here, we described how to train a support vector machine (SVM) classifier (Supplementary Figure S5). Specifically, we applied ten-fold cross-validation to obtain train data and test data. In each fold, 90% of subjects was used as a training set, and the left-out 10% subjects were used as a test set.

b) The procedure of first subtyping and then making separate models for each subtype is interesting, but it is once again optimistically biased due to leakage. Specifically, the uncertainty in subtype assignment is not accounted for. To do this properly, the authors would need to fit a new SuStaln model to the training data, and *apply* that model to the left-out data, for each fold of cross-validation. This is again to simulate what would happen in a real setting if this model were to be generalized to new data (the subjects would not already have a subtype).

Response: Thanks for the suggestion. To examine the generalization of SuStaln subtype and stage to unseen data, we conduct a ten-fold cross-validation in surgery subsample. For each fold, a new SuStaln model is trained on 90% of the cross-sectional data; the trained model is used to infer individual subtype and stage on the left-out 10% test data. We compare whether the subtype and stage assignments of unseen data are consistent with original model that has been trained on all data. It shows that 98.6% of individuals keep consistent subtype assignments with the original subtype (Supplementary Table S6). As for stage assignment, Spearman correlation test shows a high consistency between stages of unseen data and original result ($r=0.986$, $p<0.001$) (Supplementary Figure S8).

These suggest a high generalizability to unseen data. We add this generalization analysis to the revised manuscript.

In addition, the aim of this procedure is to predict post-operative outcomes. The true treatment outcome label is not leakage when ten-fold cross-validation or SuStaln modeling. The SuStaln is a data-driven unsupervised modeling based on only pre-operative image data and without any prior clinical information. For one pre-operative case who will receive surgery in the future, we could infer his/her subtype and stage using a model training on data also include his/her pre-operative data. Therefore, we hold that treatment-related information is not leakage for the procedure of first subtyping and then making separate model for each subtype.

<<The following changes have been made to the Main Text>>

In **Results 2.3 Reproducibility of SuStaln subtypes**

To examine the generalization of SuStaln subtype to unseen data, we conducted a generalization analysis with ten-fold cross-validation. For each fold, a new SuStaln model was trained on 90% of the data, and was used to infer individual subtype and stage on the left-out 10% data. We compared whether the subtype and stage assignments of unseen data are consistent with original model that has been trained on all data. We observed that 98.6% of individuals keep consistent subtype assignments with the original subtype (Supplementary Table S6). Spearman correlation test shows a high consistency between stages of unseen data and original result ($r=0.986$, $p<0.001$) (Supplementary Materials). These suggest a high generalizability of SuStaln subtype to unseen data.

<<The following changes have been made to the Supplementary Materials>>

Supplementary Table 6. Model generalizability to unseen data.

		Subtype assignment (Model generalized to unseen data)			
		Subtype 1	Subtype 2	Subtype 3	Subtype 4
Subtype assignment (original assignments)	Subtype 1	51	0	0	1
	Subtype 2	0	54	0	0
	Subtype 3	0	1	18	0
	Subtype 4	0	0	0	20

Supplementary Figure 8. Consistency of individual staging between stages of unseen data and original result ($r=0.986$, $p<0.001$, Spearman correlation test).

c) The authors don't have to do this, but to be clear, my original request was a much simpler proposition — to just add subtype (e.g. as a set of dummy variables) and stage to a prediction model, and see if this enhances prediction of treatment outcomes above and beyond a model without the subtype features.

Response: Thanks for the comments. According to the reviewer's suggestion, we conduct a logistic prediction analysis with ten-fold cross-validation. For each fold, the data of 90% subjects are used to train a logistic model to predict surgery outcome; and the left-out 10% subjects are used as test set. We compare the prediction performance for adding subtype (dummy variables) and stage to a prediction model with a model without the subtype features. Receiver operating characteristic (ROC) analysis (**Figure R2-2**) reveal that the prediction model including both clinical and subtype features exhibit higher performance (model 1, $AUC=0.621$, $p=0.017$, sensitivity of 53.1% and specificity of 74.0%, Youden Index=0.271), higher than the model only including clinical features (model 2, $AUC=0.537$, $p=0.470$, sensitivity of 87.8% and specificity of 26.0%, Youden Index=0.138). These results indicate that adding subtype feature to a prediction model do enhance prediction performance of treatment outcomes; this is consistent with our prediction results using SuStaln subtype-based classifier with ten-fold cross-validation.

Figure R2.2 Receiver operating characteristic shows the performance of prediction ability to identify individuals who are not seizure-free after surgery using logistic prediction models including subtype features (Model 1) or not (Model 2).

4) I appreciate the new analysis in extended Figure 3 showing correlations between subtype patterns. However, only same subtypes are shown. What is most useful is to show a cross-correlation across all subtypes of the discovery in validation set (as in Vogel et al., 2021 Nat Med; Figure 2B). This is necessary since a) it shows specificity and b) there is usually a baseline level of correlation between atrophy maps due to overall similarity in brain anatomy across people.

Response: Thanks for the suggestion. This is indeed a very important point. According to the reviewer's suggestion, we add extended Figure 3b showing cross-correlation across all subtypes of the discovery in validation set. It shows a low spatial correlation between two subtypes.

<<The following changes have been made to the Main Text>>

Extended Figure 3. Four distinct neuroanatomical signatures of brain atrophy patterning in people with temporal lobe epilepsy in discovery dataset and validation dataset separately. Pearson correlation coefficient is used to evaluate the consistency of z score map between discovery dataset and validation dataset.

5) R1 (comment 1) raises an excellent point stating that it isn't clear whether the atrophy patterns observed are truly "progressive" as would be suggested by the advancement of stages. The new positional variance diagrams (Fig S3) indeed show that very few subjects advance passed stage ~12. In response to this point, the authors provide an interesting and important longitudinal analysis looking at whether subtypes change over time (Fig S6). However, an analysis that would actually answer the Reviewers question more directly would be to chart change in SuStaln *stage* over time. If several subjects show advancing stage over time, this suggests the atrophy is progressive. If not, it may support R1's conjecture that perhaps the atrophy is sporadic.

Response: We thank the reviewer for suggesting us to chart change in SuStaln *stage* over time (**Figure R2.3**). We compare the SuStaln stage between the baseline and follow up for each individual. Ten of 23 individuals show progressing forward stages. The average of annual rate of SuStaln *stage* across all 23 individuals is 0.235 stage/year. This result provide partial evidence supporting the progressive atrophy.

Figure R2.3 SuStaln stage change between baseline and follow-up.

6) A new study was recently published using a very similar approach and research question to the present study (Xiao et al., 2023 Brain; <https://academic.oup.com/brain/article/146/11/4702/7300937>). The Discussion of this manuscript would definitely be improved by putting the present paper into context with this new study, to describe where findings are convergent, discrepant and complementary.

Response: We thank the reviewer's suggestion. In Xiao et al., work, their SuStaln trajectories are from mix data from several focal and idiopathic generalized epilepsies; whereas our study focuses at the data from TLE, which is the most common drug-resistant

focal epilepsy. Compared to the previous study, we have a much larger subset of individuals with TLE underwent both pre-surgical and post-surgical assessments. This allows to investigate the association of post-surgical outcomes with SuStaln subtypes.

We add a part in the Discussion to describe where findings are convergent, discrepant and complement with previous studies as follows.

<<The following changes have been made to the Main Text>>

In *Discussion*

A recent research has utilized the SuStaln algorithm to explore the progression of epilepsy-related brain atrophy [37]. The study identified different subtypes of progression, including a cortical progression subtype and a non-cortical basal ganglia subtype in both focal and idiopathic generalized epilepsies. Additionally, a third hippocampus-driven progression subtype was specifically found in focal epilepsies. This subtype involved initial volume loss in the hippocampus, followed by the thalamus, and finally affecting other cortical areas. The observed spatiotemporal trajectory in the study aligns well with the current data. A recent work [14] also confirmed a similar sequence of regional changes in people with mesial temporal lobe epilepsy and hippocampal sclerosis, through an event-based disease progression modeling [38]. These suggest that the hippocampal-dominated trajectory may be one of the most significant features in TLE people.

REVIEWERS' COMMENTS

Reviewer #2 (Remarks to the Author):

Through the authors' new analyses and rebuttal, I am now convinced that there are no longer any major methodological flaws or issues possibly confounding the results.